# Bioinspired artificial spider silk photocatalyst for the high-efficiency capture and inactivation of bacteria aerosols

Linghui Peng [1,2], Haiyu Wang [1,2], Guiying Li [1,2], Zhishu Liang [1,2], Weiping Zhang [1,2], Weina Zhao [1,2] & Taicheng An [1,2] ✉

Bioaerosol can cause the spread of disease, and therefore, capture and inactivation of bioaerosols is desirable. However, filtration systems can easily become blocked, and are often unable to inactivate the bioaerosol once it is captured. Herein, we reported a bioinspired artificial spider silk (ASS) photocatalyst, consisting of a periodic spindle structure of $TiO_2$ on nylon fiber that can efficiently capture and concentrate airborne bacteria, followed by photocatalytic inactivation in situ, without a power-supply exhaust system. The ASS photocatalyst exhibits a higher capture capacity than the nylon fiber substrate and a photocatalytic inactivation efficiency of 99.99% obtained under 4 h irradiation. We found that the capture capacity of the ASS photocatalyst can be mainly attributed to the synergistic effects of hydrophilicity, Laplace pressure differences caused by the size of the spindle knots and surface energy gradients induced by surface roughness. The bacteria captured by the ASS photocatalyst are inactivated by photocatalysis within droplets or at the air/photocatalyst interfaces. This strategy paves the way for constructing materials for bioaerosol purification.

Bioaerosols that contain living organisms, such as bacteria, viruses, pollen, spores, and fungi, remain suspended in the air for long periods of time[1–4]. They are found in both indoor and outdoor air with small sizes (1 nm–1000 μm) and varying concentrations ($1 \times 10^3$–$1 \times 10^6$ cells m⁻³), and they originate from both natural and anthropogenic activities[2,5–7]. Bioaerosol inhalation can cause serious infections and even death[8–10]. Thus, the development of bioaerosol control technology is necessary and important for air purification during the epidemic stage of COVID-19. Conventional air filters capture bioaerosols by passively blocking their motion on fibrous or porous materials via direct interception, inertial impaction, Brownian diffusion, gravity settling, and electrostatic forces[11–15]. Even though a high removal efficiency can be achieved with powerful exhaust

systems (a filtration system with a strong negative pressure), these induce large pressure drops, are easily blocked, have the low adhesive force and high energy consumption, and are unable to inactivate bioaerosols[16–19]. Therefore, it is highly desirable to find ways to overcome the above limitations and develop novel materials with integrated properties to realize both the active capture and effective inactivation of bioaerosols.

In nature, spider silk can actively capture tiny dust particles and microdroplets from air; the microdroplet coalesces to form larger droplets, which concentrate small dust particles and moisture on the spider silk[20]. Jiang et al.[21–25] reported that the water-collecting capacity of cribellate spider capture silk is the result of a unique fiber structure that consists of periodic spindle knots separated by joints. By

[1]Guangdong Key Laboratory of Environmental Catalysis and Health Risk Control, Guangdong-Hong Kong-Macao Joint Laboratory for Contaminants Exposure and Health, Institute of Environmental Health and Pollution Control, Guangdong University of Technology, Guangzhou 510006, China. [2]Guangdong Engineering Technology Research Center for Photocatalytic Technology Integration and Equipment, Guangzhou Key Laboratory of Environmental Catalysis and Pollution Control, School of Environmental Science and Engineering, Guangdong University of Technology, Guangzhou 510006, China. ✉e-mail: antc99@gdut.edu.cn

mimicking this special fiber structure, many researchers have developed artificial spider silk (ASS) for water collection from the air in arid regions[26–31]. Recently, water collection capability of ASS was improved hundreds of times by further modifying it with engineered natural silk[32], introducing a hollow structure[27], and designing a heterostructure[33]. However, all previous studies were only focused on the collection of water from the air but did not consider airborne microorganisms in bioaerosols, which mainly exist in water microdroplets, particle matter, and aggregations dispersed in air[34,35]. The bioaerosol capture capability of this versatile ASS has never been explored, let alone its underlying capture mechanism.

Photocatalytic technology shows great potential for the inactivation of biological contaminants in water due to its excellent oxidation capacity, environmental friendliness, lack of a need for external equipment, and good compatibility[36]. Shi et al. reported that effective Ag/AgX-CNT (X = Cl, Br, I) composite photocatalysts can completely inactivate $1.5 \times 10^7$ CFU mL$^{-1}$ E. coli in water within 40 min[37]. Microorganisms such as bacteria can be attacked by reactive oxygen species (ROS) generated by photocatalysts under light, which subsequently leads to cell death in water[38]. However, airborne microorganism inactivation by photocatalysts has rarely been investigated. Lu et al. reported a $TiO_2/Ti_3C_2T_x$ nanosheet-loaded polyurethane foam in a continuous flow-through reactor, and it decreased airborne E. coli by an order of 3.4-log under ultraviolet irradiation[39]. Valdez-Castillo et al.[40] developed a Perlite-supported $ZnO/TiO_2$ photocatalytic system with 70% airborne fungal/bacterial inactivation efficiency. However, upon capture by various porous materials, microorganisms usually enter the pores of these materials due to weak interactions between the microorganisms and materials. Light penetration is inhibited in the deep pores of materials; thus, photocatalytic inactivation efficiency decreases.

Herein, inspired by natural spider silk, we developed ASS with spindle knots combined with $TiO_2$ photocatalyst and assembled ASS photocatalyst arrays to capture and concentrate airborne microorganisms for in situ inactivation (Fig. 1a). The bioaerosol capture and inactivation performance of the ASS photocatalyst at different flow rates, bacterial concentrations, relative humidities (RHs) and so on were investigated. The bioaerosol capture mechanisms of ASS photocatalyst without an exhaust system were revealed by analyzing its hydrophilicity, size, and roughness of the spindle knots and joints. Further, the bioaerosol inactivation mechanisms at the interface of the air/photocatalyst and in water droplets were studied by Raman-chemiluminescence and electron paramagnetic resonance, respectively. Understanding the working mechanisms of the developed ASS photocatalyst may provide a bioaerosol control strategy that does not require an exhaust system and achieves in situ microbial inactivation under UV light irradiation, offering a solution for remediating bioaerosol contamination.

## Results

### Fabrication and characterization of ASS photocatalyst

The ASS photocatalyst was prepared by a dip coating method. As Fig. 1b shows, the nylon fibers were dipped into a polymethyl methacrylate (PMMA)/TiO$_2$ mixture and removed quickly. A thin layer of PMMA/TiO$_2$ liquid film remained attached to the fiber and then separated immediately into small droplets in the air due to Rayleigh-Taylor instability. The droplets stayed on the fiber, and the solvent was evaporated. Finally, the droplets were shrunk and dried, forming periodic spindle knot structures on the fiber (ASS photocatalyst). The prepared ASS photocatalyst is presented in Fig. 1c; equally spaced spindle knots of the same size were coated onto a fiber. The length of the joints and the width of the spindle knots were ~200 and ~180 µm, respectively. The cross-section of ASS photocatalyst shows that the circular shaped nylon fiber was fully enveloped by PMMA/TiO$_2$ with a thickness of ~30 µm, resulting in a spindle knot diameter of ~120 µm (inset of

Fig. 1c). The element mappings in Fig. 1d indicate that ASS photocatalyst was composed of C, N, O, and Ti. Among them, C, N, and O originated from the polyamide structure in the nylon fiber, while Ti was only dispersed on the spindle knots and originated from the TiO$_2$ photocatalyst.

The structure and morphology of ASS photocatalyst are described in Fig. 1e. ASS photocatalysts with diameters (2r) of 60−80 µm (measured in Supplementary Fig. 1) are shown in Fig. 1f. The structure and morphology of ASS photocatalyst were uniform when the diameter of the fiber was 60 µm, while they were incomplete and cracked when the diameter was 70 and 80 µm. This is because the small droplets could not cover the entire fiber to form a complete spindle structure[22].

The concentrations of PMMA and TiO$_2$ have substantial impacts on the structure and morphology of ASS photocatalyst. At a ratio of PMMA and solvents of 3:100 (by weight), the width (W) and height (H) of the spindle knot were 300 and 90 µm, respectively, and the shape was incomplete with large holes. When the ratio increased from 3:100 to 15:100, the spindle knots grew larger with smaller width and larger height (Fig. 1g), corresponding to 300, 250, 240, 210 µm and 90, 100, 120, 170 µm, respectively. The surface tension of the PMMA/TiO$_2$ mixture increased with increasing PMMA concentration (Supplementary Fig. 2), indicating that the droplets tended to be spherical rather than spreading out onto the fiber. Accordingly, the angle (β) of the spindle knot and length (L) of the joints increased from 31.7 to 43.6° and 200 to 300 µm, respectively (Supplementary Fig. 3). As a dispersed phase in PMMA, TiO$_2$ can significantly influence the integrity of the spindle knot structure. The spindle knot was smooth in the absence of TiO$_2$ (0:100 by weight) and became rougher with increasing TiO$_2$ concentration (0.1:100−1:100) (Fig. 1h). With the increase of TiO$_2$ concentration, more TiO$_2$ nanoparticles exposed on the surface (Supplementary Fig. 4). However, excessive TiO$_2$ addition resulted in a cracked structure for the spindle knot (5:100) due to weak interactions between TiO$_2$ and PMMA matrix. Therefore, relatively high concentrations of PMMA and TiO$_2$ facilitated the formation of an ASS photocatalyst that had a rough surface and spindle knots with large height and small width. The length of the joints increased from 200 to 500 µm when drawing speed increased from 5 to 50 cm s$^{-1}$ (Fig. 1i). A drawing speed of 90 cm s$^{-1}$ led to uneven length due to fluctuations at high speed. In addition, a horizontal drawing angle (0°) resulted in periodic spindle knots, while other angles destroyed the structure (Supplementary Fig. 5) due to gravity. A nylon fiber with a circular cross-section showed better integrity and stability than a polyester fiber and a nylon fiber with a triangular cross-section as the substrate for the ASS photocatalyst (Supplementary Figs. 6 and 7). Finally, optimal parameters for ASS photocatalyst preparation were confirmed and are presented in Supplementary Table 1. The above results suggest the successful preparation of an ASS photocatalyst containing periodic spindle knots on a fiber, and its structure is similar to wet natural spider silk[20].

### Bioaerosol capture performance of the ASS photocatalyst

The bioaerosol capture capability of the ASS photocatalyst was examined by using E. coli and B. subtilis with aerodynamic diameters of ~1.2 and 1.5 µm, respectively, as model bacteria (Supplementary Fig. 8). The optical images showed the bioaerosol capture procedures of the ASS photocatalyst (Fig. 2a). At the beginning of bioaerosol capture procedure, the bacteria present in small droplets were captured onto the joints of ASS photocatalyst (shown in Supplementary Fig. 9 and Supplementary Movie 1). In the next stage, the microdroplets on the joints grew larger and moved ~50 µm to the spindle knots within 0.05 s (Fig. 2b). Additionally, during this procedure, the neighboring microdroplets merged into larger droplets near the spindle knots (Fig. 2c). Finally, the microdroplets containing concentrated bacterial numbers aggregated and remained attached to the spindle knot of the ASS

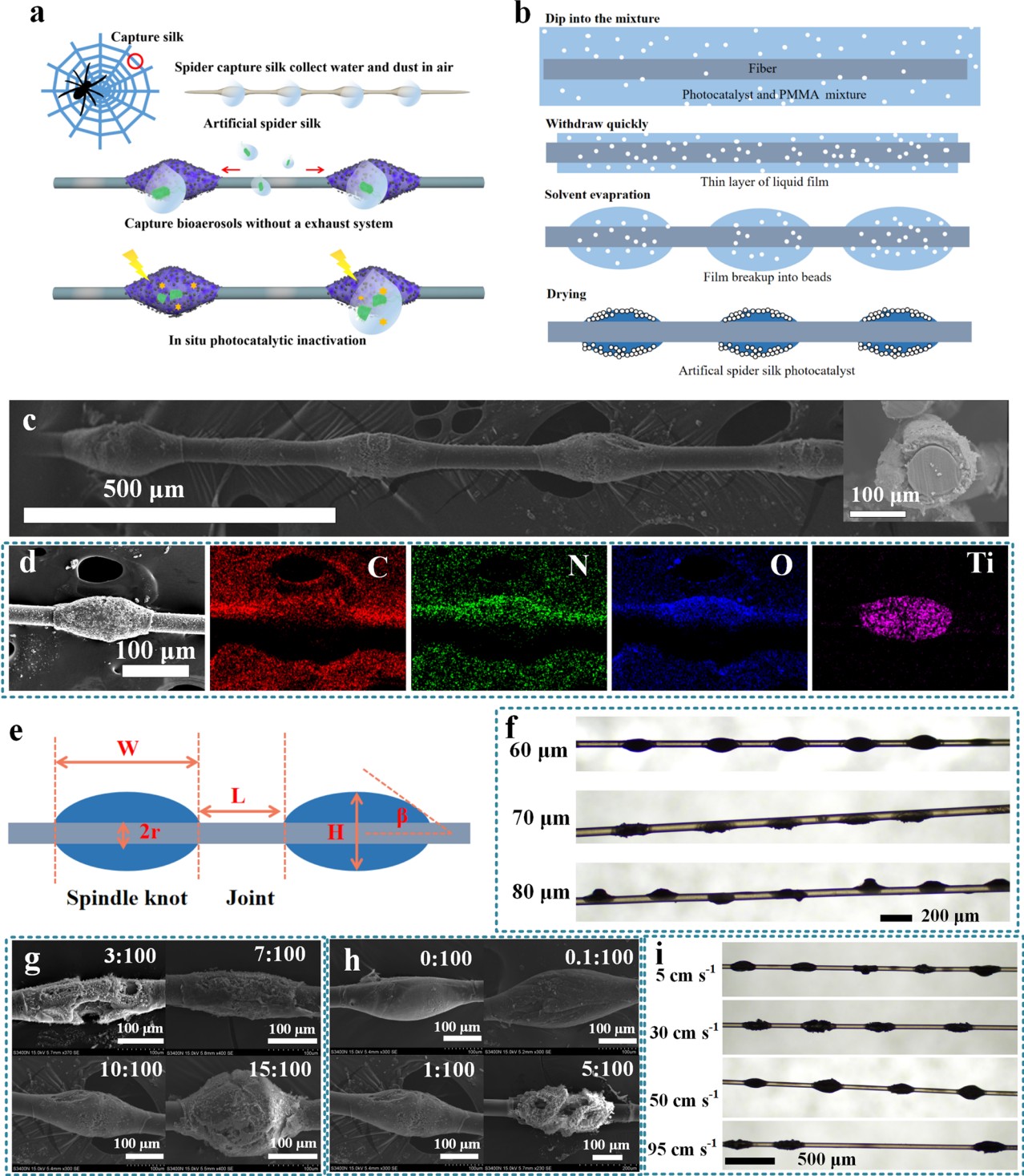

**Fig. 1 | Characterization and morphology of the ASS photocatalyst. a** Illustration of the concept of the ASS photocatalyst for capturing and inactivation of airborne bacteria. **b** Preparation processes of the ASS photocatalyst. **c** SEM images of the ASS photocatalyst and the inset is the cross-section of the ASS photocatalyst (The independent experiment has been repeated three times, and similar results were obtained). **d** EDS element mapping of the spindle knot of the ASS photocatalyst (The independent experiment has been repeated three times, and similar results were obtained). **e** Size of the ASS photocatalyst (W Width, H Height, L Length, r Radius, β Angle). **f** Digital graphs of the ASS photocatalyst with different diameters of fiber (The independent experiment has been repeated three times, and similar results were obtained). SEM images of the spindle knots with different concentrations of **g** PMMA and **h** TiO$_2$ (The independent experiment has been repeated three times, and similar results were obtained). **i** Digital graphs of the ASS photocatalyst with different drawing speeds (The independent experiment has been repeated three times, and similar results were obtained).

photocatalyst, leaving the joints exposed for the continued capture of bioaerosols.

SEM images show that bacteria were captured and concentrated within droplets on the spindle knots (Fig. 2d) rather than on the

joints. Specifically, most bacteria settled onto the surface of the spindle knots (Fig. 2e), while only a few bacteria were left on the joints (Fig. 2g). After the capture of bioaerosols, the ASS photocatalyst was cultivated. As shown in the inset of Fig. 2d, petaloid

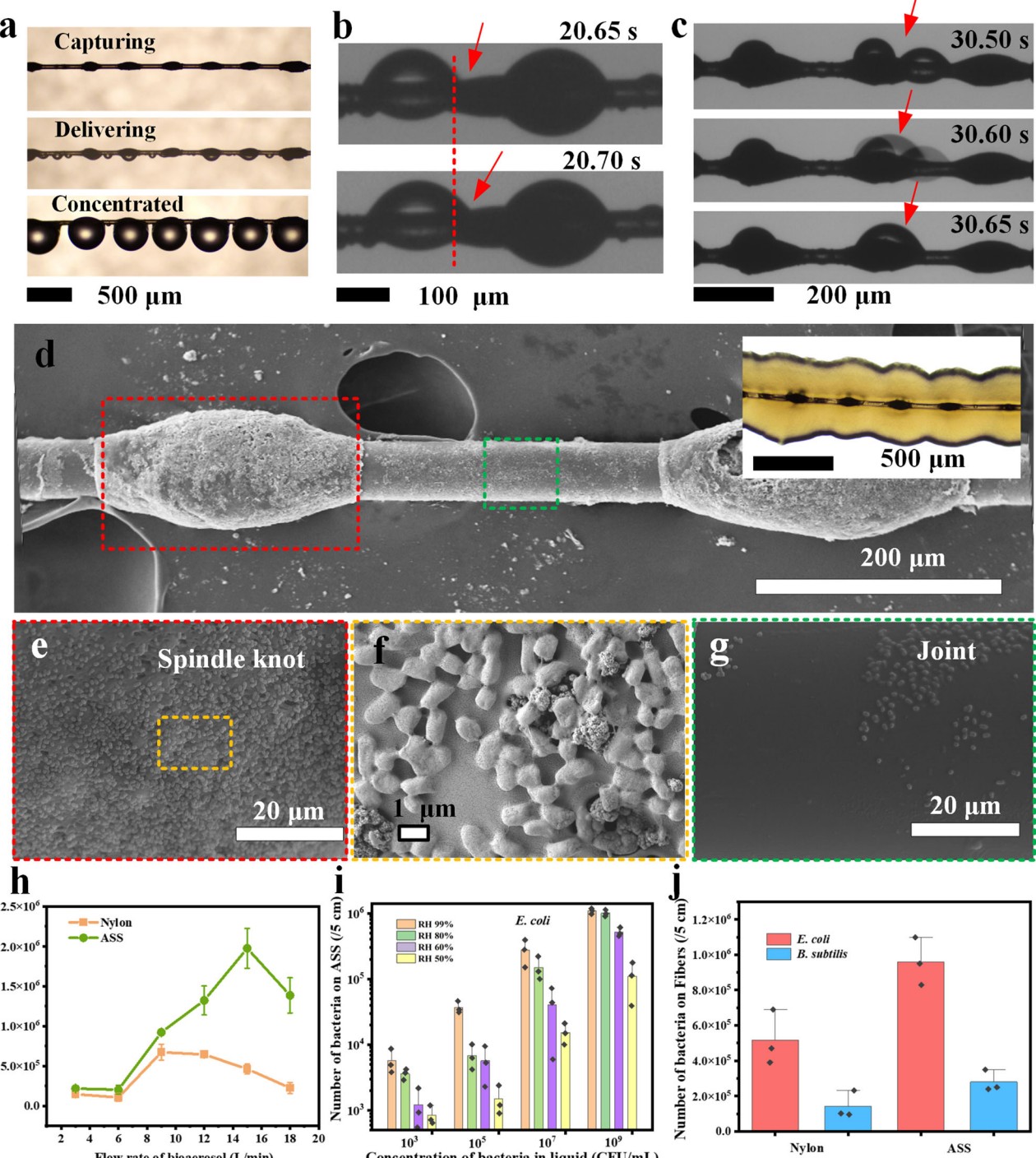

**Fig. 2 | Bioaerosols capture performance of the ASS photocatalyst. a** Optical images of bioaerosols capture processes of the ASS photocatalyst. Optical image of the droplet **b** delivers bacteria and **c** converges together towards the spindle knot of the ASS photocatalyst by a high-speed camera. **d** SEM image of the ASS photocatalyst that captured *E. coli* and the inset is optical images of bacterial colonies grown on the ASS photocatalyst after cultivation (The independent experiment has been repeated for three times, and similar results were obtained). SEM images of bacteria concentrated on the spindle knot at **e** low and **f** high magnification and **g** on the joint (The independent experiment has been repeated for three times, and similar results were obtained). Bioaerosols capture performance of the ASS photocatalyst at **h** different flow rate, **i** RH and concentration of the *E. coli* bacteria, and **j** with different types of bacteria (The error bars are calculated via repeating the measurements for three times (*n* = 3), and data are presented as mean values ± SD).

bacterial colonies formed along the spindle knots, indicating that bacteria were captured but not inactivated without light irradiation. The surface of the spindle knot was covered by *E. coli* of a short ~1 μm rod-like shape (Fig. 2f). The above results confirm that the bacteria aerosols were captured and concentrated onto the spindle knots of the ASS photocatalyst.

We investigated the quantity of bacteria captured by the ASS photocatalyst at different bioaerosol flow rates, as shown in Fig. 2h. At 3 L min⁻¹, the amount of bacteria captured by the ASS photocatalyst was slightly higher than that of a pure nylon fiber, indicating it had capture capacity in the absence of an exhaust system even at a very low flow rate. However, the capture performance of the ASS photocatalyst

was significantly better than that of pure nylon at 6−15 L min⁻¹. The peak performance of the ASS photocatalyst was obtained at 15 L min⁻¹, which was two times higher than that of pure nylon. At a low flow rate, fewer bioaerosols passed through the ASS photocatalyst over the same capture time interval; therefore, fewer bacteria were captured. With increasing flow rate, the captured microdroplets were quickly concentrated onto the spindle knots, leaving the joints as capture sites for continuous bioaerosol capture. Therefore, the ASS photocatalyst could realize a higher bioaerosol capture efficiency. However, when the flow rate was further increased (18 L min⁻¹), the bioaerosols were not easily captured and most of them just passed through along with the airflow, resulting in low captured efficiency of the ASS photocatalyst due to short retention time[41].

As shown in Fig. 2i, the number of bacteria captured onto the ASS photocatalyst decreased with decreasing concentration in the stock bacterial suspension because a high concentration ($10^9$ CFU mL⁻¹) of the stock bacterial suspension generated more bioaerosols, yielding more opportunities for interactions between the bacteria and the ASS photocatalyst. Interestingly, the bacterial concentration of the stock suspension decreased by 6 orders of magnitude (from $10^9$ to $10^3$ CFU mL⁻¹), while the number of bacteria captured onto the ASS photocatalyst decreased by only ~3 orders of magnitude. This result indicates that the ASS photocatalyst had great potential for capturing and concentrating bioaerosols at low concentrations. In addition, the number of bacteria captured onto the ASS photocatalyst increased with increasing RH. At a high RH of 99%, the ASS photocatalyst captured almost 10 times more bacteria than that at a low RH of 50%, indicating that a high RH facilitates bioaerosol capture by the ASS photocatalyst. The ASS photocatalyst can also capture bioaerosols with different aerodynamic diameters, namely, *E. coli* and *B. subtilis*, capturing 2 times more of these bacteria than pure nylon (Fig. 2j). The above results confirm the bioaerosol capture capacity of the ASS photocatalyst under different conditions.

## Bioaerosol capture mechanism of the ASS photocatalyst

According to the results above, we assumed that the high bioaerosol capture performance of the ASS photocatalyst could be attributed to the hydrophilicity of the ASS photocatalyst. We compared the bioaerosol capture performance of the ASS photocatalyst on nylon fiber substrate and polyester fiber substrate (Supplementary Fig. 10), because the hydrophily of these fibers is different. Because the amide groups present on nylon fibers are hydrophilic[42], the ASS photocatalyst on nylon fiber substrate could capture 5 and 1.5 times as many *E. coli* and *B. subtilis* cells as on hydrophobic polyester fiber substrate, respectively (Fig. 3a). Furthermore, Fig. 3b shows that the water contact angle ($\theta$) of the joint at RH 50% was 97.5° ($\theta > 90°$, hydrophobic), while at RH 80%, it was 88.9° ($\theta < 90°$, hydrophilic)[22,43]. For the spindle knots, $\theta$ was 125.3° at RH 50% and decreased to 93.6° at RH 80%, showing hydrophobicity under both conditions. The results indicates that the hydrophilicity of the ASS was improved at high RH. At both high and low RH levels, the $\theta$ of the spindle knot was higher than that of the joint under the same conditions. Atomic force microscopy (AFM) was also applied to measure the adhesive force between the bacteria and the surface of the ASS photocatalyst to reveal the bioaerosol capture mechanism. The top of Fig. 3c illustrates that the bacterial probe (~20 μm, Supplementary Fig. 11) interacted with the spindle knots and the joints on the ASS photocatalyst at different RHs. As shown in the bottom of Fig. 3c, the average adhesive forces between the *E. coli* bacteria and the surface of the spindle knots and joints were 8.4 and 9.0 nN for RH 50% and 25.3 and 28.1 nN for RH 80%, respectively. For *B. sublitis*, the adhesive forces of the spindle knot and the joint were 23.3 and 27.6 nN for RH 50%, and 32.6 and 40.4 nN for RH 80%, respectively. At low RH, the adhesive force was significantly lower than that at high RH maybe due to less hydrogen bonds between water molecules adsorbed on the ASS and the bacteria. The theoretical

calculation of hydrogen bonds between the ASS and bacteria shows that the interaction energy between bacteria and the joint and spindle knot are 0.93 and 0.8 eV, respectively at low RH, while are enhanced to −5.32 and −4.09 eV, respectively at high RH (calculated in Supplementary Fig. 12). In addition, the liquid film may form on the ASS due to hydrophily at high RH, thus capillary force may contribute to the adhesive forces between bacteria and the ASS[44] (representative adhesive force curves are shown in Supplementary Fig. 13). The adhesive forces of the bacteria with the joints were higher than those with the spindle knots at different RHs, indicating that the bioaerosols had affinity for the joints. Therefore, the bioaerosols preferred the joints, and the first step in bacterial capture from bioaerosols by the ASS photocatalyst was an adhesive force that was driven by hydrophilicity between the joints and bacteria.

Due to the hydrophilicity of the joints of the ASS photocatalyst, it has affinity for bioaerosols, but the final numbers of bacteria captured on the ASS photocatalyst were closely related to the morphology of the ASS photocatalyst. We prepared ASS photocatalysts with different height (H), length (L), and β values (namely samples 1, 2, 3, and 4), and the related parameters of these samples are listed in Supplementary Table 2. The volumes of bacterial droplets collected on the ASS photocatalysts of different height, length and β values are shown in Fig. 3d. The calculated droplet volumes of samples 3 and 4 were similar (~1 μL) but larger than those of samples 1 and 2 (~0.2 μL) (Supplementary Table 3), indicating that large spindle knots were beneficial for efficiently capturing microdroplets from air. SEM images also revealed that more bacteria were captured by the ASS photocatalyst that had large spindle knots and small length (Supplementary Figs. 14 and 15). The number of bacteria captured by the ASS photocatalyst increased with the size of the spindle knots and decreased with an increase in the length of the joints (Fig. 3e), corresponding to the results of the droplet volume captured, as shown in Fig. 3d.

To determine the force driving behind the movement of bacterial droplets from the joints to spindle knots, the interaction between the droplets and the surface of the ASS photocatalyst was further investigated. Surface energy gradients, which can arise from differences in either surface chemical composition (hydrophilic and hydrophobic property shown in Fig. 3b) or surface roughness, could offer driving forces for droplet movement. From the AFM and SEM images (Fig. 3f), we found that the surface of the joints was relatively smooth, had a roughness of 7.0 nm, contained very few particles. On the other hand, the surface of the spindle knots was rough due to exposed TiO₂ particles on its surface and due to DMF evaporation (which resulted in porous and rough surfaces), which resulted in a roughness of 176 nm. However, in the absence of TiO₂, the roughness was 126 nm. The difference in surface roughness between the joints and the spindle knots was ~170 nm, which was slightly higher than that without TiO₂ (~120 nm). This difference in surface roughness between the spindle knots and joints resulted in surface energy gradients, causing differences in the advancing angle and receding angle of the droplet. As shown in Fig. 3g, the water contact angle on the spindle knot was 71.5° (advancing angle), and for the joint, it was 106.7° (receding angle). That is, the spindle knot was easier to wet and had a higher apparent surface energy than the joint. Even though the spindle knot was more hydrophobic than the joints due to the chemical composition shown in Fig. 3b, the droplets captured onto the ASS photocatalyst preferred to wet the spindle knots with higher apparent surface energy, leading to movement away from the joints and towards the spindle knots[20,22]. The bioaerosol capture performance of the ASS photocatalyst with TiO₂ was slightly higher than that without TiO₂ addition for both *E. coli* and *B. subtilis* in the bioaerosols (Fig. 3h), which confirms the finding that a larger surface roughness difference facilitates bioaerosol capture.

The proposed mechanisms for ASS photocatalyst bioaerosol capture and concentration are illustrated in Fig. 3i, and it includes hydrophilicity, a Laplace pressure difference caused by the size of the

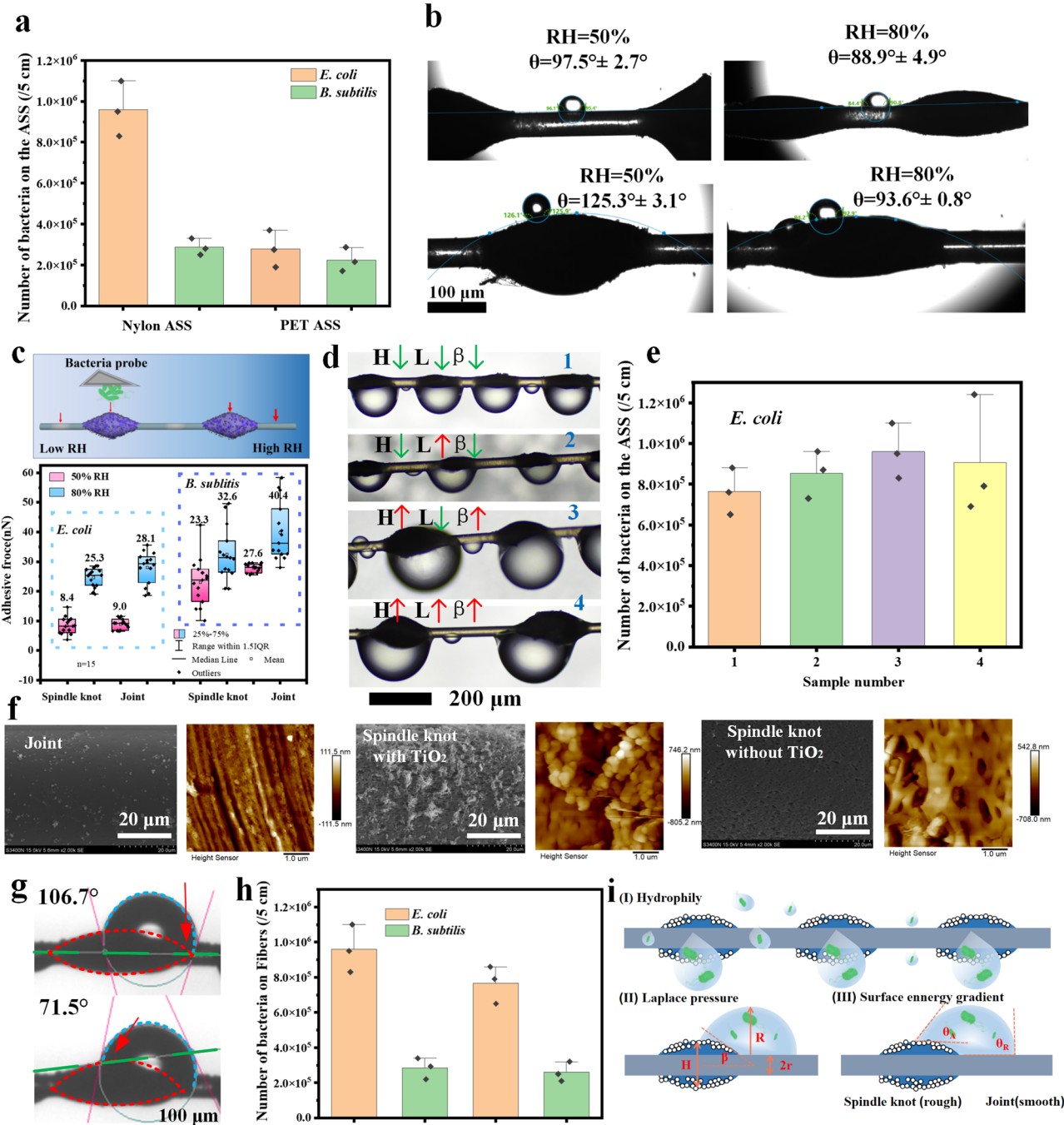

**Fig. 3 | Bioaerosols capture mechanisms of the ASS photocatalyst. a** Bioaerosols capture performance of the ASS photocatalyst with different fiber substrate (The error bars are calculated via repeating the measurements for three times (*n* = 3) and data are presented as mean values ± SD). **b** Water contact angles of the single ASS photocatalyst at different RH (The independent experiment has been repeated for three times, and similar results were obtained). **c** Adhesive force between the bacteria and the ASS photocatalyst at different RH (The independent experiment has been repeated 15 times, and similar results were obtained, *n* = 15). **d** Optical images of captured bioaerosols by the ASS photocatalyst with different β, height (H) of the spindle knots, and length (L) of the joints. **e** Bioaerosols capture performance of the ASS photocatalyst with different morphology (The error bars are calculated via repeating the measurements for three times (*n* = 3), and data are presented as mean values ± SD). **f** SEM images and AFM images of the ASS photocatalyst (the independent experiment has been repeated for three times, and similar results were obtained). **g** Advancing and receding angles of the droplet on the ASS photocatalyst. **h** Bioaerosols capture the performance of the ASS photocatalyst with and without TiO$_2$ (The independent experiment has been repeated for three times, and similar results were obtained, and data are presented as mean values ± SD). **i** Illustration of the proposed bioaerosols capture and concentration mechanisms of the ASS photocatalyst (*H* height, *R* radius of the droplet, *r* radius of the fiber, *β* angle of the spindle knot, $θ_A$ advancing angle, $θ_R$ receding angle. The error bars are calculated via repeating the measurements for three times (*n* = 3)).

spindle knots and surface energy gradients generated by the surface roughness. The hydrophilic property of the ASS photocatalyst induces their strong affinity for bioaerosols (that exist as microdroplets and hydrophilic solid particles); thus, it captures more hydrophilic bacteria

(Fig. 3i (I)). The driving force for bioaerosol concentration and movement arises from the spindle-shaped geometry of the spindle knots, which generates differences in Laplace pressure ($\triangle$P described in SI) (Fig. 3i (II)). The Laplace pressure difference between the joints and the

high curvature site (the joint with local radius r) is larger than that on the low curvature site (the spindle knot with local radius H/2). Because the diameter of the fiber is smaller than the height of the spindle knot, the resultant nonequilibrium Laplace pressure difference within a water drop propels droplets containing bacteria from the joint to the spindle knot[20]. Then, the droplets grow large and make contact with the spindle knot and the joint of the ASS photocatalyst, forming $\theta_A$ (advancing angle) and $\theta_R$ (receding angle), respectively. As shown in Fig. 3i (III), the spindle knot is composed of $TiO_2$ nanoparticles, while the joint is smooth nylon fiber, therefore, the spindle knot region has a higher roughness than joint region, which gives rises to a driving force generated by a surface energy gradient to move the droplet towards the spindle knot (described in SI)[20]. The mechanism proposed above suggests that the highly hydrophilic properties of the ASS photocatalysts, the large spindle knots/relatively short joints (Laplace pressure differences caused by the size of the spindle knots), and the large surface roughness differences (surface energy gradients induced by surface roughness) facilitate bioaerosol capture. These driving forces synergistically enable the ASS photocatalyst to actively capture bioaerosols without the use of a powerful exhaust system. Notably, after the droplets mobilize bacteria onto the spindle knots and leave more hydrophilic sites (joints) vacant and exposed for further bioaerosol capture, a new cycle of bioaerosol capture, concentration, and movement can begin at the joints.

## Bioaerosol inactivation performance of the ASS photocatalyst

The bacterial droplets present on the spindle knots of the ASS photocatalyst can provide a photocatalytic microenvironment for in situ and continuous bacterial inactivation by ROS generated under UV light photocatalysis. Light yellow *E. coli* bacterial colonies grew along with the ASS photocatalyst without UV light photocatalysis (Fig. 4a), while no noticeable bacterial colonies were seen after UV irradiation of the ASS photocatalyst for 4 h (Fig. 4b), indicating that the bacteria captured by the ASS photocatalyst were inactivated by photocatalysis. The bottom of Fig. 4a, b show the same results. Before photocatalysis, the bacteria on the ASS photocatalyst were intact and had rod-like shapes and smooth surfaces (eluted bacteria grew before irradiation (insets)), but after photocatalysis, the bacteria became small, flaccid, flat and cracked (no colonies were observed). *B. subtilis* bacteria have 2 μm rod-like shapes and smooth surfaces. After UV light photocatalysis, the cells shrank and damaged (holes formed on the cell body) (Fig. 4c). Eluted *B. subtilis* bacteria also grew into colonies before light irradiation, while no colonies were observed after UV light photocatalysis. The results confirm that the bacteria captured onto the ASS photocatalyst could be completely inactivated under UV light photocatalysis.

In Fig. 4d, we found that the inactivation efficiency of the ASS photocatalyst increased with increasing $TiO_2$ loading under UV light irradiation. This result can be easily explained by the fact that a greater quantity of $TiO_2$ results in more ROS production, and the ROS interact with the bacteria, leading to higher inactivation efficiency. However, as mentioned in Fig. 1h, to balance the integrity of spindle knots and the high inactivation efficiency, we applied a concentration of 1 g/100 mL $TiO_2$ to the stock PMMA/$TiO_2$ mixture to prepare the ASS photocatalyst.

Notably, the number of living bacteria on the ASS photocatalyst first decreased slowly and then quickly with increasing light irradiation time at both low ($10^3$ CFU mL$^{-1}$) and high ($10^9$ CFU mL$^{-1}$) concentrations of stock bacteria (Fig. 4e). During the first hour of irradiation, the bacteria and ASS photocatalyst were present in the water droplets, which weakened UV light absorption and protected the bacteria from dehydration; thus, the inactivation efficiency during the first hour was relatively low (evaporation of the droplet shown in Supplementary Fig. 16). With continued irradiation, the bacteria made contact with the $TiO_2$ on the spindle knots after the droplets dried, producing both

dehydration and more opportunities for ROS interactions with the bacteria, leading to rapid and high-efficiency inactivation. The bacteria captured on the ASS photocatalyst could be completely inactivated (from ~$10^4$ to 0 CFU/5 cm) after irradiation for 4 h (stock bacteria is $10^3$ CFU mL$^{-1}$). In fact, 99.99% inactivation efficiency was achieved at $10^9$ CFU mL$^{-1}$ of stock bacteria. Clearly, intracellular ROS levels of *E. coli* bacteria increased significantly after UV light photocatalysis (Fig. 4f), indicating that the bacteria were injured by oxidative stress. The *E. coli* bacterial cell activities decreased with prolonged irradiation time (Fig. 4g), showing that the bacteria were attacked and inactivated under photocatalysis. The above results demonstrate the effective photocatalytic inactivation performance of the ASS photocatalyst.

The photocatalytic inactivation performance of the ASS photocatalyst increased with increasing UV light intensity regardless of the type of bacteria (Fig. 4h). Both the *E. coli* and *B. subtilis* bacteria were almost completely inactivated (inactivation efficiency >99.99%) by the ASS photocatalyst within 4 h at an irradiation intensity of 15 mW cm$^{-2}$. The $C/C_0$ of *B. subtilis* was slightly smaller than that of *E. coli*, indicating *B. subtilis* could be inactivated more easily. Recent studies have evidenced that the cell inactivation by photocatalyst is due to the attack of ROSs, leading to membrane and cell wall damage[36]. The Gram-negative bacteria (*E. coli*) have a complex cell wall and additional outer membrane-lipopolysaccharide, comparing with the Gram-positive bacteria (*B. subtilis*). Thus, Gram-negative bacteria (*E. coli*) can protect themselves from the attack of ROSs to a certain extent[45,46].

As shown in Fig. 4i, when the RH decreased from 99 to 50%, the percentage of living bacteria on the ASS photocatalyst after 4 h of photocatalysis decreased from 0.8% to ~0%, indicating that the bacteria dehydrated and inactivated more rapidly at low RH. The results indicate that the gram-negative/positive bioaerosols captured by the ASS photocatalyst were inactivated under UV light photocatalysis, and the inactivation efficiency increased with increasing irradiation intensity and decreasing RH.

## Photocatalytic inactivation mechanism of the ASS photocatalyst

We found that after bacteria were captured by the ASS photocatalyst, at first, they could be slowly inactivated in situ on the spindle knots and then could be quickly destroyed under UV light irradiation. Thus, we proposed the following two modes of photocatalytic inactivation mechanisms: photocatalytic inactivation in water droplets and at the interface of the spindle knots and air (Fig. 5a). In the droplets, the ROS generated by the ASS photocatalyst under UV light photocatalysis were at the water-solid interface. Figure 5b shows the electron paramagnetic resonance (EPR) spectra of DMPO-•$O_2^-$, DMPO-•OH, TEMP–$^1O_2$, and TEMPO-h$^+$ measured in aqueous solution with the ASS photocatalyst under UV light irradiation or in the dark. Abundant holes (h$^+$), singlet oxygen ($^1O_2$), hydroxyl radicals (•OH) and superoxide radicals (•$O_2^-$) were detected in water under UV light irradiation, indicating that ROS diffused into the water to attack dispersed bacteria. Among them, the strong signals of h$^+$ and •OH indicated that they were potentially the dominant contributors to bacterial inactivation. In addition, the results also indicated that the $TiO_2$ particles on the spindle knots were exposed to UV light to produce ROS and were not covered by PMMA. These EPR spectra confirm that the ASS photocatalyst generated ROS under UV light irradiation to inactivate bacteria in water droplets. After the water droplets evaporated, the bacteria that adhered to the surfaces of the spindle knots were inactivated by the ROS generated at the interface between the spindle knots and air. To determine the fate of the surface of the ASS photocatalyst in air under UV light irradiation, we applied a Raman technique to record the chemiluminescence emission spectra of luminol, which can detect weak signals from samples[47,48]. When luminol reacts with ROS, it can generate blue fluorescence (~420 nm), which can be used to signal the generation of ROS by the ASS photocatalyst under UV light irradiation. As demonstrated in

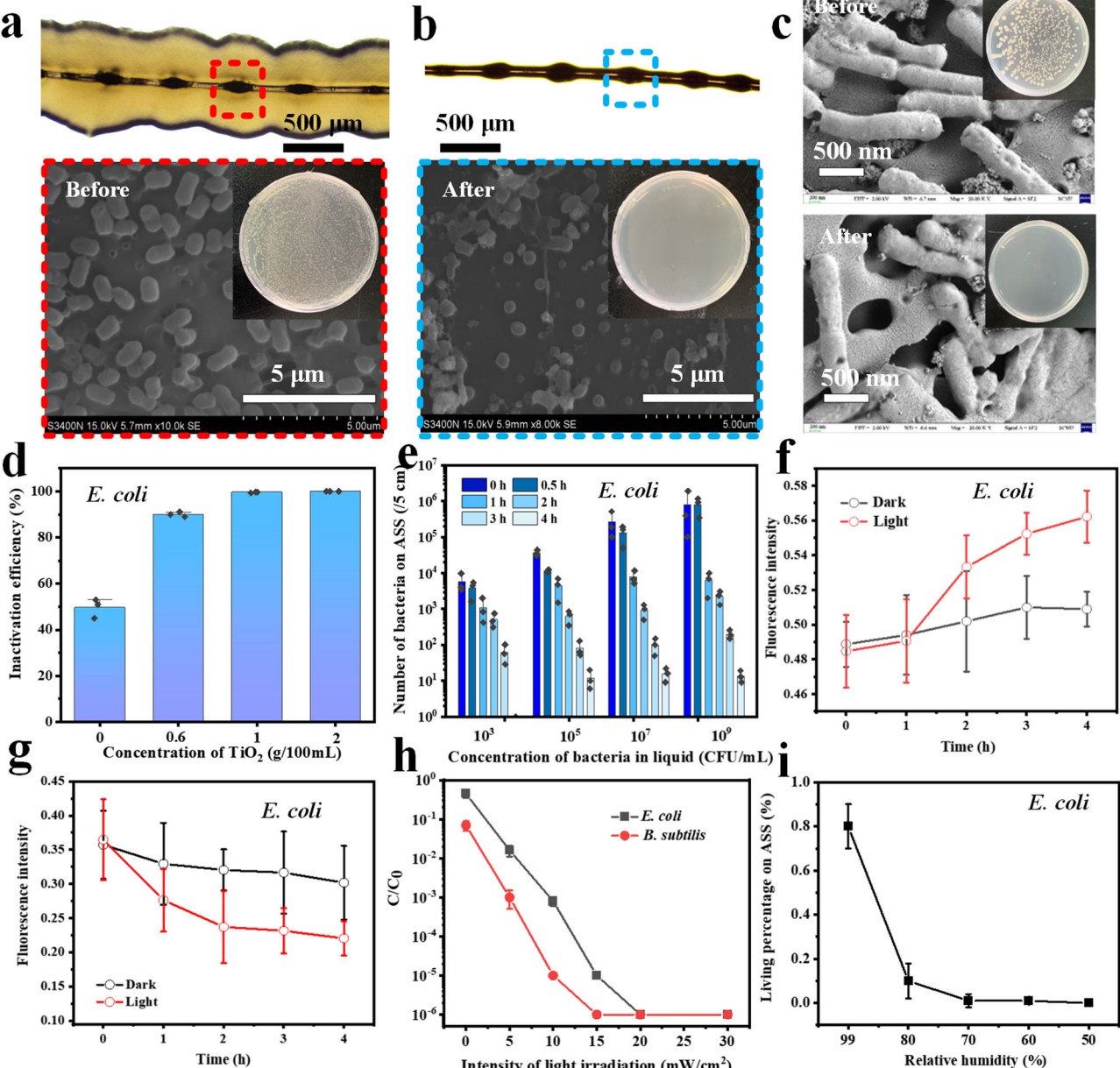

**Fig. 4 | Bioaerosols inactivation performance of the ASS photocatalyst.** Optical and SEM images of the ASS photocatalyst after *E. coli* bioaerosols capture cultivated on nutrient medium **a** before and **b** after UV light irradiation (The independent experiment has been repeated for three times, and similar results were obtained). **c** SEM and optical images of *B. subtilis* captured by the ASS photocatalyst before and after irradiation (The independent experiment has been repeated for three times, and similar results were obtained). Inactivation efficiency of the ASS photocatalyst **d** with different TiO₂ loading and **e** at different irradiation time and concentration of bacteria (The error bars are calculated via repeating the measurements for three times ($n = 3$), and data are presented as mean values ± SD). **f** Intracellular ROS and **g** cell viability of *E. coli* bacteria captured on the ASS photocatalyst for different irradiation time (The independent experiment has been repeated for three times, and similar results were obtained, and data are presented as mean values ± SD). Inactivation efficiency of the ASS photocatalyst **h** under different intensity of light irradiation, and **i** at different RH (The error bars are calculated via repeating the measurements for three times ($n = 3$), and data are presented as mean values ± SD).

Fig. 5c, the bulk of the ASS photocatalyst was irradiated for 5 min under a UV laser, and then luminol was dropped onto the ASS photocatalyst to immediately record its emission spectrum. We found that the characteristic peak at 420 nm increased significantly relative to samples without irradiation or luminol when laser irradiation was applied to luminol and the ASS photocatalysts (Fig. 5d). These results confirm that ROSs were generated at the interface of the photocatalyst and air under UV light irradiation, which was previously reported by Akira Fujishima[49,50]. These ROS generated in the water or the interface attacked captured bacteria, disrupting their cytomembranes and cell walls to some extent (in Fig. 4b, c)[16]. This result suggests that bacteria are inactivated by the ASS photocatalyst via two modes: first by attacking from ROSs in the water droplets and then at the interface of air and photocatalyst. It is worth mentioning that the ASS photocatalyst can be more fully exposed to light irradiation for photocatalysis than other filters that are based on exhaust systems due to its active capture capacity. In addition, since airborne bacteria are concentrated onto spindle knots, the ROSs that generated on the spindle knots are likely to have more opportunities to interact with bacteria for inactivation. Therefore, the high inactivation efficiency of the ASS photocatalyst is

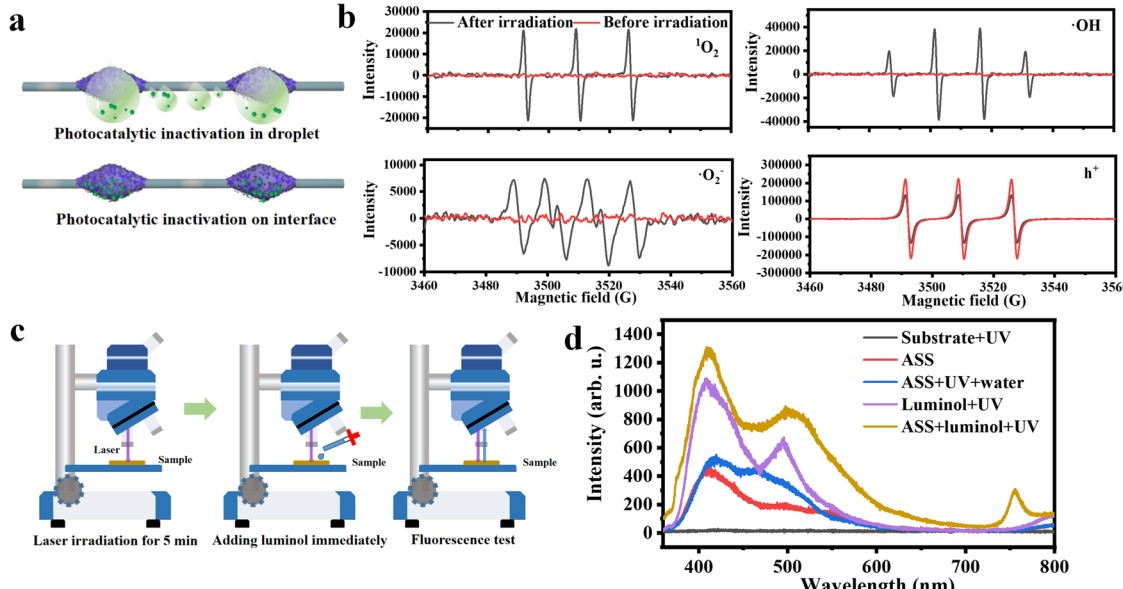

**Fig. 5 | Bioaerosols inactivation mechanisms of the ASS photocatalyst.**
**a** Illustration of the two states of the bacteria on the ASS photocatalyst. **b** EPR spectra of $^1O_2$, $h^+$, $O_2^-$ and $\cdot OH$. **c** Illustration of the Raman-chemiluminescence test of the ASS photocatalyst. **d** Chemiluminescence spectra of luminol dropping on the ASS photocatalyst before and after light irradiation, the ASS photocatalyst in interface and water, and comparisons of the luminol, the ASS photocatalyst and the combination.

attributed to the concentration of bacteria on the spindle knots and ROSs generation at the droplet/interface under UV light irradiation.

## Discussion

In summary, we developed an ASS photocatalyst by integrating $TiO_2$ with periodic spindle knots and investigated the capture and inactivation performances of bioaerosols as well as their corresponding mechanisms in detail. The bioaerosol capture performance of the ASS photocatalyst was two times higher than that of pure nylon, and an inactivation efficiency of 99.99% was achieved. Bioaerosols were first captured by the hydrophilic joints, and then they moved towards the spindle knots, leaving the hydrophilic capture sites exposed for further bioaerosol capture. The bioaerosol capture mechanism of the ASS photocatalyst was as follows: airborne bacteria were actively captured in droplets and driven in a controllable direction by optimizing the curvature (Laplace pressure difference), chemical (hydrophilic), and roughness gradients (surface energy gradients) onto the surfaces of the ASS photocatalyst. The concentrated bacteria on the spindle knots were inactivated in situ under UV light irradiation. Moreover, ROS generated by the ASS photocatalyst was shown to exist in the water and at the interface between the photocatalyst and air, indicating that photocatalytic inactivation processes could occur both in the water and at the photocatalyst-air interface. This study provides a strategy for bioaerosol active capture and in situ photocatalytic inactivation without the use of an exhaust system.

## Methods
### Materials and characterization
Nylon fibers, titanium dioxide (P25, Supplementary Fig. 17), and poly-methyl methacrylate (PMMA) were purchased from Macklin Co., Ltd., and the other materials used in this study are listed in the Supporting Information (SI). Characterization methods such as scanning electron microscopy (SEM), electron paramagnetic resonance (EPR), and atomic force microscopy (AFM) are described in SI.

### Preparation of the ASS photocatalyst
A fiber with mixed $TiO_2$/PMMA spindle knots was prepared by immersing a nylon fiber in a $TiO_2$/PMMA/(DMF + ethanol) solution and

drawing it out at a velocity of $5 - 95\,cm\,s^{-1}$. A thin film formed on the fiber surface and spontaneously separated into periodic polymer droplets along the fiber due to the Rayleigh instability and then dried in air. Periodic photocatalyst spindle knots formed on the nylon fiber (artificial spider silk called the ASS), and the geometry was similar to that of the wetted capture silk of spiders.

### Capture performance measurement of the ASS photocatalyst
Gram-negative *E. coli* (K-12) and gram-positive *B. subtilis* (ATCC6633) were used as model bacteria aerosols in this study. Detailed experimental processes were described in the SI, and the experimental setup was also illustrated in Supplementary Fig. 18. The assembled ASS photocatalysts (four 5-cm arrays) were placed in the generated bioaerosol flow field for 2 min to capture airborne bacteria. The bacteria captured by the photocatalyst were eluted and plated for counting.

### Photocatalytic inactivation performance of the ASS photocatalyst
First, bioaerosols were captured by the ASS photocatalyst, and then the photocatalyst was placed under UV light irradiation (365 nm) for 0 to 4 h. After irradiation, the ASS photocatalysts were immersed in 20 mL saline solutions in an ultrasonic bath for 20 min, and the solution was vortexed for 1 min and then diluted for plating. The number of colonies was enumerated through visual inspection. The calculation method for inactivation efficiency is described in the SI. All experiments were repeated three times.

### Reporting summary
Further information on research design is available in the Nature Portfolio Reporting Summary linked to this article.

## Data availability
The additional data are provided in the Supplementary Information. All the data sets generated and analyzed during the current study are available from the corresponding authors on request. Source data are provided with this paper.

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

## Acknowledgements

This work was supported by the National Natural Science Foundation of China (U1901210 receipted by G.L. and 42207112 receipted by L.P.), Science and Technology Project of Guangdong Province, China (2021A0505030070 receipted by G.L. and 2022A1515010538 receipted by L.P.), Local Innovative and Research Teams Project of Guangdong Pearl River Talents Program (2017BT01Z032, receipted by T.A.).

## Author contributions

Conceptualization: L.P., T.A.; methodology: L.P., T.A., H.W., G.L., Z.L., W.Z.; investigation: L.P., T.A., H.W., G.L, Z.L., W.Z.; funding acquisition: L.P., G.L., T.A.; project administration: L.P., T.A.; supervision: T.A.; writing, original draft: L.P., T.A.; writing, review, and editing: H.W., G.L., Z.L., W.Z. modeling: W.N.Z.

## Competing interests

The authors declare no competing interests.
