## [Peer Review File · Nature Communications]

Bioinspired Artificial Spider Silk Photocatalyst for the High-efficiency Capture and Inactivation of Bacteria AerosolsREVIEWER COMMENTS

Reviewer #1 (Remarks to the Author):

Bioaerosol inactivation is becoming an emergent requirement due to the outbreak of COVID-19. This study has developed a novel system which combined the capture and inactivation of bioaerosol into one photocatalyst system. This novel system was inspired by the spider silk web. High capture capacity and inactivation efficiency were demonstrated. And the corresponding mechanisms were also investigated, presented and discussed. I would like to recommend the publication of this manuscript after minor revision. Some comments are given below:

Major comments:

1. Since the work showed promising results in capturing and inactivating the bioaerosols, it might be better if the authors could provide a more detailed information regarding the ASS photocatalyst setup for bioaerosol capture and inactivation. This would provide a future application possibility of this novel ASS photocatalyst. From the methods section, it could be seen that four ASS photocatalysts were assembled into an array. How were they assembled?
2. How was the RH controlled in the chamber? Based on information from the SI, it seemed that the different flow rates (3–18 L/min) were used to adjust the RH. What kind of exposure chamber was used for the bioaerosol capture by the ASS array?
3. What did the “without a power-supply exhaust system” mean? As mentioned in the above point, different flow rates were applied.
4. How long will the droplet last before its total evaporation? It seemed that the inactivation efficiency was low at the droplet mode, and increased rapidly with the UV irradiation.
5. Abstract: It’s mentioned that the synergistic effects of hydrophilicity, Laplace pressure differences and surface energy gradients accounted for the ASS capture capacity. The authors might need to provide more detailed discussion about the synergistic effects.
6. How about the size of the microdroplet attached on the fiber? For real cases, particles in the air are generally in a more solid form with less water. Will this affect the capture capacity?

Minor points:

Line 99: The first name of the word was used hereafter. Since the words here like width, height, length are not very long, it might be better to use them directly instead of the first capital. For example, “L” in the sentence (line 113) seemed a bit weird for the author to understand. Similarly, it might be better

with an attributive word before the “2r” in the sentence in line 243.

Line 121: it should be “an ASS”

Line 198: Is a word “joint” missing in the first sentence “Due to the hydrophilicity of the ASS photocatalyst”? From the last paragraph, it seems that the hydrophilicity is between the joint and the bacteria.

Line 203-204: The meaning of the samples 1, 2, 3, 4 should be made clear here.

Figure 2: What kind of bacteria was used for panel i? Moreover, a different color combination might be better in panel j. Since the adjacent of the panel I and j, the reader would possibly assume the colors means the same thing.

Reviewer #2 (Remarks to the Author):

Authors reported a bioinspired artificial spider silk (ASS) photocatalyst consisting of a periodic spindle structure of TiO₂ onto nylon fibre that can efficiently capture and concentrate airborne bacteria and photocatalytically inactivate them in situ without a power-supply exhaust system. In general, the idea of this work is good but of less novelty (similar work has been published, Water Research, 2021, 206: 117759.). There is a lot of experimental data, however, conclusions as authors stated can not be drawn based on present data. Some questions need also to be addressed as followings:

1. In the coating of PMMA/TiO₂@ nylon fibres, how can authors conform whether the TiO₂ nano particles are fully covered by PMMA? Does the percentage of coverage influence the efficiency of photocatalysis? Detailed explanation is required.

2. In figure 1 d, I'm wondering whether the Ti mapping describe the usual case (as the figure presented, Ti only presented on spindle knot rather than the joint).

3. “However, when the flow rate was further increased (18 L min⁻¹), the bioaerosols were not easily captured due to inertia.” When the flow rate was increased to 18 L min⁻¹, the bioaerosols capture performance is rather worse. Why inertia have an effect on it? Are there any references?

4. As presented in the manuscript, RH can affect the bioaerosols capture performance. How to control the experimental RH was not introduced in Experimental section.

5. “At low RH, the adhesive force was significantly lower than that at high RH due to a lack of hydrogen bonds and the surface tension of water at the wetted surface (representative adhesion force curves are shown in Supplementary Fig. 11).” Detailed explanation of lack of hydrogen bonds should be given. High RH can enhance the capillary condensation between bacterial and the testing surface. Liquid bridge

needs a large force to detach.

6. All the Contact Angle data presented is not in the form of average data ($CA \pm ?$, at least three times' repeating), so less the credibility. More seriously, the CAs on the knot is $\sim 126^\circ$, the water advancing and receding contact angle on knot are respectively $\sim 106^\circ$ and 71° , how can contact angle larger than the advancing contact angle? further explanation is required.

7. " $\theta_A < \theta_R$ means that the droplet prefers to spread towards the spindle knot rather than towards the joints, generating a driving force for droplet movement from the joint to be concentrated onto the spindle knots (described in SI)", why $\theta_A < \theta_R$ means that? Are there any references? The mechanism of droplet self-propelling should be reconstructed (how surface wettability difference generates directional driving force).

Reviewer #3 (Remarks to the Author):

This manuscript reports the photocatalytic bacterial inactivation of a bioinspired artificial spider silk (ASS) made from a periodic spindle structure of TiO_2 on a nylon fiber. ASS showed the capture capacity of the bioaerosols twice as high as the pure nylon fiber and 99.99% inactivation efficiency on both Gram-negative *E. coli* bacteria and Gram-positive *B. subtilis* bacteria under UV light. The authors attributed its capture mechanism to the synergistic effect from the hydrophilicity, size of the spindle knots, and surface roughness. The concentrated bacteria on the spindle knots are inactivated by ROS generated through the photocatalytic reactions. The results provide a very interesting strategy that can be adopted for the purification system of the bioaerosols without using the high-power air exhaust system.

The authors termed the hydrophobicity as a water contact angle (θ) $> 90^\circ$ and the hydrophilicity as that $< 90^\circ$, so the surface with $\theta = 95.8^\circ$ was hydrophobic while that with $\theta = 87.6^\circ$ was hydrophilic. But both seem to be hydrophobic to me. Such a criterion may mislead the surface characteristics.

Also, they used AFM with a bacterial probe to measure its adhesion with the surface of the spindle knots and the joints. But they did not identify the type of the bacteria used for an AFM probe. Isn't the surface interaction dependent on the type of the bacteria on the AFM probe?

There have been quite a few reports on ASS for its design and structure with potential applications, but its use in the photocatalytic purification process is an interesting idea by creating the periodic spindle structure of well-known photocatalytic oxide (TiO_2). While the structural design and processing scheme seem to be well executed, some explanations on the resultant microstructure details are missing and bacteria killing mechanisms are not clear. For example, it creates the periodic fiber spindle knots made of the TiO_2 – PMMA cluster, but detailed representation of such spindle knot surfaces was not given in the manuscript so it is not clear how surface morphologies play a role in the effective photocatalytic inactivation of bacteria. TiO_2 nanoparticles need to be

exposed to the surface to be effective, so microstructure characterization of the hybrid arrangements of TiO₂ in PMMA resulting from its various concentrations should be discussed in relation to the corresponding antibacterial properties. In Fig. 4 (d), the bacterial inactivation efficiency is shown to increase with TiO₂ loadings (i.e., more ROS production) until it reaches to a certain concentration of TiO₂. In order to relate the origin of the saturation, microstructure evolution at the top surface of the TiO₂ – PMMA mixture needs to be explained. Degussa P25 is well-known TiO₂ nanoparticles, but it still needs to report the range of particle sizes of TiO₂ in the manuscript.

Bacterial inactivation mechanism needs some clarification too. The authors stated that the cell wall of the Gram-negative bacteria (*E. coli*) is thinner than that of the Gram-positive bacteria (*B. subtilis*), making the former more easily inactivated by ROS. However, the Gram-negative bacteria has an additional outer membrane (which is not present for the Gram-positive bacteria), so they are in general more resistant to anti-biotics and causing more serious diseases for human. They may be more difficult to inactivate with the ROS generated from TiO₂ photocatalysts as well. Furthermore, the author's data do not warrant their claim either. Therefore, it may not be attributed to the blockage of light absorption due to more *E. coli* bacteria on fibers.

The testing results need to be compared with the control sample, but this study provides no data from the bacterial viability testing with the pure nylon fiber or the ASS catalysts without TiO₂. This is because these bacteria can be killed under UV light or after the droplets dry up on the control sample surfaces. Further, in the Results and Conclusion sections, the author claimed 99.99% efficiency (4 log reduction) while it was 99.999% (5 log reduction) in the Abstract. This discrepancy needs to be corrected.

Figure 5 (d) is confusing. It was supposed to show that more ROS were generated at the surface of ASS in air or in the water droplet captured on ASS, but the two highest intensity cases were 'ASS/luminol' and 'Luminol irradiated'. I am not sure how the label of these samples is related to the aforementioned two cases (at the surface of ASS in air; inside the water droplet with the ASS photocatalyst).

Photocatalytic inactivation properties of ASS were measured under the UV light with a wavelength of 365 nm. But the ROS generation experiments were done with the UV light with a wavelength of 325 nm. This discrepancy needs to be explained to make the relevance between two experiments.

In the References section, only a single author is included in each of the reference article. Normally, full author names should be included in the references.

In summary, this paper presents a novel approach to mimic a spider silk by using a periodic spindle structure of TiO₂ on a nylon fiber. The processing schemes are well designed to capture the bioaerosols that contain bacteria. But there seemed to be no clear microstructural description of TiO₂ nanoparticles spread in PMMA and their influence on antibacterial performances under UV light. In addition, some of the experimental techniques are not well coordinated each other,

so it requires more explanation. Also, there were no data provided from control sample for the bacterial viability test. Inactivation mechanism of the bacteria via photocatalytic reactions needs to be better explained or confirmed with the data to support the author's conclusion on the Gram-negative vs. the Gram-positive bacteria.

Response to the comments for NCOMMS-22-48373

Title: Bioinspired Artificial Spider Silk Photocatalyst for the High-efficiency Capture and Inactivation of Bacteria in Bioaerosols

REVIEWER COMMENTS

Reviewer #1 (Remarks to the Author):

Comments: Bioaerosol inactivation is becoming an emergent requirement due to the outbreak of COVID-19. This study has developed a novel system which combined the capture and inactivation of bioaerosol into one photocatalyst system. This novel system was inspired by the spider silk web. High capture capacity and inactivation efficiency were demonstrated. And the corresponding mechanisms were also investigated, presented and discussed. I would like to recommend the publication of this manuscript after minor revision. Some comments are given below:

Response: Many thanks for your highly positive to our work. We have further improved our manuscript according to reviewer's comments.

Major comments:

Question 1. Since the work showed promising results in capturing and inactivating the bioaerosols, it might be better if the authors could provide a more detailed information regarding the ASS photocatalyst setup for bioaerosol capture and inactivation. This would provide a future application possibility of this novel ASS photocatalyst. From the methods section, it could be seen that four ASS photocatalysts were assembled into an array. How were they assembled?

Response: Thanks for the useful comments, and we have added the experimental setup and ASS photocatalyst array in the Manuscript and also SI. As shown in Supplementary Fig. 18, the experimental setup consisted of two major parts, including bioaerosol generation part and capture & inactivation part. The bioaerosol with different RHs

was sprayed into the Teflon box (188 L) for subsequent experiments.

On the other hand, four ASS photocatalysts with 5 cm long were fixed on the holder parallelly, standing in front of the bioaerosol spraying in the box (illustrated in Supplementary Fig. 18). The UV lamp was placed on the back of the ASS array.

Author's changes in Manuscript: Page 19, Line 401-402, “Detailed experimental processes were described in the SI, and the experimental setup was also illustrated in Supplementary Fig. 18.”

Author's changes in SI: Please see Page 2-3, Line 28-32, “The experimental setup is consisted of two major parts, including bioaerosol generation part, and capture & inactivation part. The bioaerosol with different RHs was sprayed into the Teflon box (188 L) for later experiments. Four ASS photocatalysts with 5 cm long were fixed on the holder parallelly, standing in front of the bioaerosol spraying in the box. The UV lamp was placed on the back of the ASS array.”

Please see Page 15 in SI for Figure.

Supplementary Fig. 18. Illustration of experimental setup

Question 2. How was the RH controlled in the chamber? Based on information from the SI, it seemed that the different flow rates (3–18 L/min) were used to adjust the RH. What kind of exposure chamber was used for the bioaerosol capture by the ASS array?

Response: Thanks for the considerable question. As shown in newly-supplied Supplementary Fig. 18, the RH in the chamber was controlled by injecting bioaerosol with different RHs adjusting by the diffusion dryer (circled by red dashed line).

Supplementary Fig. 18. Illustration of experimental setup

The diffusion dryer (PERMA PURE Nafion MD-700) with a length of 120 cm is a suitable RH controller for aerosol drying based on the moisture diffusion with less bioaerosol loss. The diffusion dryer is a concentric tube, showing in the following pictures. The bioaerosols go through the central tube, and moisture is diffused into purge gas with different flow rate or vacuum degree due to moisture gradients, yielding bioaerosols with different RHs. Thus, the RHs of the injecting air or bioaerosol can be tuned by adjusting the air flow rate or vacuum of the purge gas in shell area. The chamber is rinsed by the bioaerosols for 30 min, thus RH in chamber can be changed by injecting bioaerosols with different RHs.

Picture 1 Illustration of diffusion dryer

Picture 2 Picture of diffusion dryer

The exposure chamber is a Teflon film chamber with 188 L to reduce wall loss of bioaerosol. Thus, we have made some changes in the SI accordingly.

Author's changes in SI: Please see Page 3, Line 36-42, “The diffusion dryer (PERMA PURE Nafion MD-700) with a length of 120 cm is selected for bioaerosol drying based

on moisture diffusion. The bioaerosols go through the central tube and moisture is diffused into purge gas or vacuum (shell tube) due to moisture gradients, yielding bioaerosols with different RHs. Thus, the RHs of the injecting air or bioaerosol can be tuned by adjusting the air flow rate or vacuum of the purge gas in shell area. The chamber is rinsed by the bioaerosols for 30 min, thus RH in chamber can be changed by injecting bioaerosols with different RHs.”.

Question 3. What did the “without a power-supply exhaust system” mean? As mentioned in the above point, different flow rates were applied.

Response: In our study, we characterized the bioaerosol capture and inactivation performance of the ASS without using a filtration system with a strong exhaust system (negative pressure), because only few ASS fibers arrays were used rather than fiber mats. The captures of bioaerosols by ASS materials with adsorbing and concentrating performance rather than filtration. Thus, the exhaust system for filtration was not suitable in this system for this ASS strategy. As shown in Supplementary Fig. 18, the bioaerosol was not filtered by the ASS and we didn’t use a power-supply exhaust system at the end of outlet. Moreover, the different flow rates of bioaerosol gas come from the bioaerosol generation system.

Supplementary Fig. 18. Illustration of experimental setup

Author's changes in Manuscript: Please see Page 2, Line 31: “Even though a high removal efficiency can be achieved with powerful exhaust systems (a filtration system with a strong negative pressure), these induce large pressure drops, are easily blocked, have low adhesive force and high energy consumption, and they are unable to

inactivate bioaerosols¹⁷⁻²⁰”.

Please also see Page 19, Line 401: “Detailed experimental processes were described in the SI, ...”

Author's changes in SI: Please see Page 3, Line 32-35: “The bioaerosol capture and inactivation performances of the ASS photocatalyst were characterized without using a filtration system with a strong exhaust system (negative pressure) at the end of outlet. The bioaerosol flow rates come from the bioaerosol generation system.”

Question 4. *How long will the droplet last before its total evaporation? It seemed that the inactivation efficiency was low at the droplet mode, and increased rapidly with the UV irradiation.*

Response: Thanks for your nice question. We roughly monitored the droplet evaporation at different RH (15 °C) when irradiated. We have added the time of droplet evaporation in the Manuscript and Supplementary Figure 16.

Author's changes in Manuscript: Page 14, Line 299-300, “thus, the inactivation efficiency during the first hour was relatively low (evaporation of the droplet shown in Supplementary Fig. 16).” .

Author's changes in SI: Please see Page 15 for Figure.

Supplementary Fig. 16. Time of droplet evaporation at different RHs (15 °C)

Question 5. *Abstract: It's mentioned that the synergistic effects of hydrophilicity, Laplace*

pressure differences and surface energy gradients accounted for the ASS capture capacity. The authors might need to provide more detailed discussion about the synergistic effects.

Response: Thanks for your comprehensive comments. We have added detailed discussion about the synergistic effects in the manuscript.

Author's changes in Manuscript: Page 13, Line 263-268, “The mechanism proposed above suggests that the highly hydrophilic properties of the ASS photocatalysts, the large spindle knots/relatively short joints (Laplace pressure differences caused by the size of the spindle knots), and the large surface roughness differences (surface energy gradients induced by surface roughness) facilitate bioaerosol capture. These driving forces synergistically enable the ASS photocatalyst to actively capture bioaerosols without the use of a powerful exhaust system.” .

Question 6. *How about the size of the microdroplet attached on the fiber? For real cases, particles in the air are generally in a more solid form with less water. Will this affect the capture capacity?*

Response: The size of the microdroplet attached on the fiber is ~100-200 μm in diameter (shown in Fig. 3d). The average volume of the droplet is around 0.5 μL that calculating by spherical volume formula ($V=1/2*4/3\pi R^3$), and the volume of droplets hang on the ASS (Fig. 3d) is listed in Supplementary Table 3.

As shown in Fig. 2i, the capture capacity decreases with decrease of RH. In a real environment, water molecules should be existed in bioaerosols even in very dry environment. Thus, the ASS photocatalyst can also work, but with just a lower performance.

Fig. 2i Number of bacteria captured on the ASS with different RH

Minor points:

Question 1. Line 99: The first name of the word was used hereafter. Since the words here like width, height, length are not very long, it might be better to use them directly instead of the first capital. For example, “L” in the sentence (line 113) seemed a bit weird for the author to understand. Similarly, it might be better with an attributive word before the “2r” in the sentence in line 243.
Line 121: it should be “an ASS”

Response: Thanks for your careful comments. We have revised the related abbreviations and mistakes according to your suggestion throughout the manuscript.

Question 2. Line 198: Is a word “joint” missing in the first sentence “Due to the hydrophilicity of the ASS photocatalyst”? From the last paragraph, it seems that the hydrophilicity is between the joint and the bacteria.

Response: Thanks for your kind reminder, we have revised as following.

Author's changes in Manuscript: Please see Page 10, Line 209-211, “ Due to the hydrophilicity of the joints of the ASS photocatalyst, it has affinity for bioaerosols, but the final numbers of bacteria captured on the ASS photocatalyst were closely related to the morphology of the ASS photocatalyst.”

Question 3. Line 203-204: The meaning of the samples 1, 2, 3, 4 should be made clear here.

Response: Thanks for kind comments. We have marked as follows.

Author's changes in Manuscript: Please see Page 10, Line 211-213, “We prepared ASS photocatalysts with different height (H), length (L) and β values (namely samples 1, 2, 3 and 4), and the related parameters of these samples are listed in Supplementary Table 2.”

Question 4. *Figure 2: What kind of bacteria was used for panel i? Moreover, a different color combination might be better in panel j. Since the adjacent of the panel I and j, the reader would possibly assume the colors means the same thing.*

Response: Thanks for your kind reminder. The *E. coli* was used in Fig 2i, and also marked in the following figure in Fig 2i. The color of Fig. 2j has also changed accordingly as suggested by you.

Author's changes in Manuscript:

Fig. 2. Bioaerosols capture performance of the ASS photocatalyst. (a) Optical images of bioaerosols capture processes of the ASS photocatalyst. Optical image of the droplet (b) delivers bacteria and (c) converges together towards to the spindle knot of the ASS photocatalyst by a high-speed camera. (d) SEM image of the ASS photocatalyst that captured *E. coli* and the inset is optical images of bacterial colonies grown on the ASS photocatalyst after cultivation. SEM images of bacteria concentrated on the spindle knot at (e) low & (f) high magnification and (g) on the joint. Bioaerosols capture performance of the ASS photocatalyst at (h) different flow rate, (i) RH and concentration of the *E. coli* bacteria, and (j) with different types of bacteria.

Reviewer #2 (Remarks to the Author):

Comments: Authors reported a bioinspired artificial spider silk (ASS) photocatalyst consisting of a periodic spindle structure of TiO_2 onto nylon fibre that can efficiently capture and concentrate airborne bacteria and photocatalytically inactivate them *in situ* without a power-supply exhaust system. In general, the idea of this work is good but of less novelty (similar work has been published, *Water Research*, 2021, 206: 117759.). There is a lot of experimental data, however, conclusions as authors stated can not be drawn based on present data. Some questions need also to be addressed as followings:

Response: Thanks for your critical comments. We have carefully checked the referenced data and compared your mentioned work (*Water Research*, 2021, 206: 117759) together with our work reported herein. We think that the novelty of our work is strongly different from the reference. It is because that although different biomimetic functional materials were employed in both systems, our study mainly works on airborne bacteria capture and inactivation in atmosphere, and clarifies the underlying mechanisms of bioaerosol capture and inactivation. However, the reference mainly applied in the water collection from air and purification of pesticides in water interface. As the idea of our work is quite different from the work (*Water Research*, 2021, 206: 117759), we believed the idea in this work is not only good than the reference but also novelty. The detailed reasons are also presented hereafter:

Firstly, the mechanisms of the bioaerosol capture and inactivation by the ASS photocatalyst is novel. Generally, air filters capturing bioaerosols by passively filtration (exhaust system) was reported early without inactivation capability. Our work is the first time to develop an ASS photocatalyst inspired by natural spider silk for bioaerosol capture and inactivation. The ASS photocatalyst is not just based on filtration mechanism, but the affinity and absorption of bioaerosols onto ASS photocatalyst. In addition, the captured airborne microorganisms can be concentrated and inactivated *in situ* to prevent secondary contamination.

Secondly, the synthesized material of our work is completely different from the reference (*Water Research, 2021, 206: 117759.*). They investigated cactus- and beetle-inspired water collection by patterning sharp hydrophilic triangles on the 2D plane hydrophobic surfaces, which is different from our 3D artificial spider silk with spindle knots and joints materials in our work. Not only the hydrophily of the ASS, but also the special curve, size and gaps of the spindle knot drives micro-droplets (microorganism containing) moving and capture. In addition, the bioaerosols contain water and airborne bacteria, thus the bioaerosol capture and inactivation by the ASS photocatalyst are different from water collection, which have never been attempted yet.

Thirdly, the research focus of these two works is different. The reference focuses on the performance of water directional collection and purification inspired by the *nature*, which has been investigated early. However, we focused on the mechanisms and performance of bioaerosol capture and inactivation by our newly-synthesized artificial spider silk photocatalyst, which have not been investigated yet. The understanding of underlying mechanisms of bioaerosol capture and *in situ* inactivation by the ASS photocatalyst is still quite challenge for the real application.

Finally, the application of these two works is different. The reference majorly investigates wastewater purification polluted with pesticide in water interface, while our study focused on the airborne bacteria capture and inactivation in atmosphere, which are completely different applications. Moreover, the way of interaction between photocatalyst and organics is different from that with microorganism, thus the design of photocatalyst should also be different. Furthermore, comparing with polluted water in liquid phase in the reference, the bioaerosol is a complex multi-phase system. They contain liquid and solid phases in the gaseous phase. Therefore, the bioaerosol capture is more complex and challengeable. In addition, the flow rate, RH, concentration and type of airborne bacteria in atmosphere had also been considered in our study for better application, which is closer to indoor air purification as well as bacteria inactivation rather than pesticide degradation in water interface in the reference.

Detailed comments

Question 1. *In the coating of PMMA/TiO₂@ nylon fibres, how can authors conform whether the TiO₂ nano particles are fully covered by PMMA? Does the percentage of coverage influence the efficiency of photocatalysis? Detailed explanation is required.*

Response: Thanks for your useful comments. The SEM images of the ASS at high magnification show that the TiO₂ nanoparticles are exposed on the surface of the spindle knot when the concentration of TiO₂ in PMMA is suitable, like 1 g/100 mL TiO₂ (Supplementary Fig. 4). In addition, the EPR spectra (Fig. 5b) of the ASS photocatalyst with and without light irradiation also tell the truth that the TiO₂ exposed on the surface. The generation of ROS indicates that TiO₂ can absorb UV light on the surface rather than covered by PMMA.

Fig. 5. Bioaerosols inactivation mechanisms of the ASS photocatalyst. (b) EPR spectra of ¹O₂, h⁺, O₂^{·-} and ·OH.

We have added the detailed surface morphology of the ASS with different TiO₂ concentration in Supplementary Fig. 4. It can be seen that with the increase of TiO₂ concentration in the PMMA solution, more TiO₂ nanoparticles exposed onto the surface. As discussed in the manuscript (Fig. 4d), the inactivation efficiency of the ASS photocatalyst can be influenced by the TiO₂ loading. Detailed discussion can be found in the manuscript as following.

Please see Page 14, Line 288-294. “In Fig. 4d, we found that the inactivation efficiency of the ASS photocatalyst increased with increasing TiO₂ loading under UV light irradiation. This result can be easily explained by the fact that a greater

quantity of TiO₂ results in more ROS production, and the ROS interact with the bacteria, leading to higher inactivation efficiency. However, as mentioned in Fig. 1h, to balance the integrity of spindle knots and the high inactivation efficiency, we applied a concentration of 1 g/100 mL TiO₂ to the stock PMMA/TiO₂ mixture to prepare the ASS photocatalyst.”

To describe the surface morphology and TiO₂ exposure on the spindle knot of the ASS photocatalyst more clearly, in addition to Supplementary Fig. 4, we also revised the statement in the manuscript to accordingly.

Author's changes in Manuscript: Page 6, Line 110-111. “With increase of TiO₂ concentration, more TiO₂ nanoparticles exposed on the surface (Supplementary Fig. 4).”

Author's changes in SI:

Supplementary Fig. 4 Surface morphology of the ASS with different TiO₂ concentrations in PMMA

Question 2. *In figure 1d, I'm wondering whether the Ti mapping describe the usual case (as the figure presented, Ti only presented on spindle knot rather than the joint).*

Response: Ti element mapping used in the manuscript is not a special case. We also did an additional element mapping of the other sample as shown in following Figure. It can be seen that the Ti element mapping in the Figure is similar with the Ti mapping in Fig. 1d. In the Figure, very weak signals also can be seen on the joints for Ti

element. Only very few Ti exists on the joints due to surface tension of PMMA during spindle knot formation. The TiO₂ nanoparticles concentrated along with liquid PMMA into spindle knots due to Rayleigh-Taylor instability. Therefore, TiO₂ nanoparticles are majorly concentrated on the spindle knot rather than the joint of the ASS photocatalyst.

Figure. Additional element mapping of the ASS

Fig. 1. Characterization and morphology of the ASS photocatalyst. (d) EDS element mapping of the spindle knot of the ASS photocatalyst.

Question 3. “However, when the flow rate was further increased (18 L min^{-1}), the bioaerosols were not easily captured due to inertia.” When the flow rate was increased to 18 L min^{-1} , the bioaerosols capture performance is rather worse. Why inertia have an effect on it? Are there any references?

Response: Thanks for your instructive question. We are sorry that the statement of “However, when the flow rate was further increased (18 L min^{-1}), the bioaerosols were not easily captured due to inertia.” is not accurate enough. We have revised the statement and cited a reference as follows.

Author's changes in Manuscript: Please see Page 8, Line 156-159, “However, when the flow rate was further increased (18 L min^{-1}), the bioaerosols were not easily captured and most of them just passed though along with the airflow, resulting in low captured efficiency of the ASS photocatalyst due to short retention time⁴².”

Reference, please see Ref.42. Wang, H., et al. Photocatalytic ozonation inactivation of bioaerosols by NiFeOOH nanosheets *in situ* grown on nickel foam. *Appl. Catal. B: Environ.*

Question 4. As presented in the manuscript, RH can affect the bioaerosols capture performance.

How to control the experimental RH was not introduced in Experimental section.

Response: Thanks for your kind reminder. As shown in Supplementary Fig. 18, the RH in the chamber was controlled by bioaerosol with different RHs controlled with the diffusion dryer (circled by red dashed line).

Supplementary Fig. 18. Illustration of experimental setup

The diffusion dryer (PERMA PURE Nafion MD-700) with a length of 120 cm is a suitable RH controller for aerosol drying based on moisture diffusion with less bioaerosol loss. The diffusion dryer is a concentric tube, showing in the following pictures. The bioaerosols go through the central tube, and moisture is diffused into purge gas with different flow rate or vacuum degree due to moisture gradient, yielding bioaerosols with different RHs. Thus, the RHs of the injecting air or bioaerosol can be tuned by adjusting the air flow rate or vacuum of the purge gas in shell area. The chamber is rinsed by the bioaerosols for 30 min, thus RH in chamber can be changed by injecting bioaerosols with different RHs.

Picture 1 Illustration of diffusion dryer

Picture 2 Picture of diffusion dryer

The exposure chamber is a Teflon film chamber with 188 L to reduce wall loss of bioaerosol. Thus, we have made some changes in the SI accordingly.

Author's changes in SI: Please see Page 3, Line 38: “The diffusion dryer (PERMA PURE Nafion MD-700) with a length of 120 cm is selected for bioaerosol drying based on moisture diffusion. The bioaerosols go through the central tube and moisture is diffused into purge gas or vacuum (shell tube) due to moisture gradients, yielding bioaerosols with different RHs. Thus, the RHs of the injecting air or bioaerosol can be tuned by adjusting the air flow rate or vacuum of the purge gas in shell area. The chamber is rinsed by the bioaerosols for 30 min, thus RH in chamber can be changed by injecting bioaerosols with different RHs. ”

Question 5. “At low RH, the adhesive force was significantly lower than that at high RH due to a lack of hydrogen bonds and the surface tension of water at the wetted surface (representative adhesive force curves are shown in Supplementary Fig. 11).” Detailed explanation of lack of hydrogen bonds should be given. High RH can enhance the capillary condensation between bacterial and the testing surface. Liquid bridge needs a large force to detach.

Response: Thanks for your thoughtful comments. To figure out the contribution of hydrogen bonds to adhesive force under different relative humidity, we further conducted the theoretical calculation of hydrogen bonds between the ASS and the airborne bacteria as the proof of the results in Fig. 3c.

To provide insights into the adhesive force under the interested conditions of low RH or high RH, we have calculated the adsorption geometrical structure of one water molecule and then found the most favorable adsorption structure of H₂O on the joint (Figure S12a)/spindle knot (Figure S12c) from several possible

configurations. At low RH, less water molecules adsorbed on the ASS and bacteria. It clearly shows that water adsorbs on the joint via one H atom forming a hydrogen bond with one N atom of the joint (Nylon, polyamide), while there is also a hydrogen bond of water with one O atom of the spindle knot (TiO₂, P25). In accordance with the experiment results (Fig. 3c), the interaction energy of water molecular on the joint was obtained as -0.93 eV with the distance H---NH of ~1.9 Å at the low RH, higher than that of water molecular on the spindle knot of -0.80 eV with the distance H---OH of ~2.3 Å. Under high RH conditions, more water molecules adsorbed on the ASS and bacteria, and we found that the other water molecules near to the adsorbed water interact with the adsorbed water via hydrogen bonding, forming a liquid bridge on the joint (Figure S12b). Similarly, a weakly-bound water layer was also found on the spindle knot (Figure S12d), but with a less hydrogen bonds and far distance above the surface. In consist with the structural change, we found that the interaction between water molecules on both the joint and spindle knot are enhanced to -5.32 and -4.09 eV, respectively, which is contributed to the more hydrogen bonds. From the above discussion, it clearly indicates that at both low RH and high RH, the adhesive force on the joint was stronger than that on the spindle knot. At high RH, the adhesive force was significantly stronger than that at low RH due to more hydrogen bonds.

Supplementary Fig. 12. Theoretical model of the local hydrogen bonds: at the low (a) and high (b) RH situations on the joint, at the low (c) and high (d) RH situations on the spindle knot. The dark brown, light brown, blue, white and red stick balls are C, Ti, N, H and O atoms, respectively. The dotted green line indicates the hydrogen bonds.

In addition, we also agree with the reviewer's comment that the capillary condensation at high RH may contribute to the adhesive forces between bacteria and the ASS. At high RH, the liquid film may form on the ASS due to hydrophilicity, thus capillary force may contribute to the adhesive forces between bacteria and the ASS. Therefore, in order to express more clearly, we added the discussion of hydrogen bonds in the manuscript & SI and also added the statement to point out the capillary forces in the manuscript.

Author's changes in Manuscript: please see Page 10, Line 196-204, “At low RH, the adhesive force was significantly lower than that at high RH maybe due to less hydrogen bonds between water molecules adsorbed on the ASS and the bacteria. The theoretical calculation of hydrogen bonds between the ASS and bacteria shows that the interaction energy between bacteria and the joint & spindle knot are 0.93 and 0.8 eV, respectively at low RH, while are enhanced to -5.32 and -4.09 eV, respectively at high RH (calculated in Supplementary Fig. 12). In addition, the liquid film may form on the ASS due to hydrophilicity at high RH, thus capillary force may contribute to the adhesive forces between bacteria and the ASS⁴⁵ (representative adhesive force curves are shown in Supplementary Fig. 13).”

Reference: Ref. 45. Busnaina, A. A. et al. The effect of relative humidity on particle adhesion and removal. *J. Adhesion*, **74** (1-4), 391-409 (2000).

Author's changes in SI: Please see Page 4-5, Line 73-89 of SI, “**Molecular computational details.** To investigate the hydrophilicity mechanism on the joint and spindle knot, the density functional theory (DFT) calculations were performed in this work by DMol3 module in Materials Studio. The generalized gradient approximation (GGA) with the correction by the Perdew-Burke-Ernzerhof (PBE) was used as the exchange-correlation functional. The core electrons were calculated by all electron method and double numerical plus polarization (DNP) is applied as the basis set with the orbital cut off of 3.9 Å in our calculations. The convergence tolerance of energy was set to 10^{-5} Hartree (1 Hartree = 27.21 eV), and the maximum allowed

force and displacement were 0.004 Hartree/Å and 0.005 Å, respectively. The smearing parameter of 0.005 Ha was taken to facilitate structure convergence. A slab model of titanium dioxide is used while a vacuum space of ~13 Å is taken to avoid the interactions between periodic slabs. The interaction energy (ΔE_{int}) of water molecular on joint and spindle knot is defined by the following equation:

$$\Delta E_{\text{int}} = E(\text{n} \cdot \text{H}_2\text{O}/\text{sur}) - E(\text{n} \cdot \text{H}_2\text{O}) - E(\text{sur})$$

where $E(\text{n} \cdot \text{H}_2\text{O}/\text{sur})$ was the total energy of species/surface complex, $E(\text{sur})$ and $E(\text{n} \cdot \text{H}_2\text{O})$ were the total energies of corresponding adsorption carrier and the sole water moleculars, respectively. The more negative value from this equation indicated the stronger interaction accordingly.”.

Author's changes in SI: Please see Page 12, Line 169-187: “To provide insights into the adhesive force under the interested conditions of low RH or high RH, we have calculated the adsorption geometrical structure of one water molecule and then found the most favorable adsorption structure of H₂O on the joint (Figure S12 (a))/spindle knot (Figure S12 (c)) from several possible configurations. At low RH, less water molecules adsorbed on the ASS and bacteria. It clearly shows that water adsorbs on the joint via one H atom forming a hydrogen bond with one N atom of the joint (Nylon, polyamide), while there is also a hydrogen bond of water with one O atom of the spindle knot (TiO₂, P25). In accordance with the experiment results (Fig. 3c), the interaction energy of water molecular on the joint is obtained as -0.93 eV with the distance H--NH of ~1.9 Å at the low RH, higher than that of water molecular on the spindle knot of -0.80 eV with the distance H--OH of ~ 2.3 Å. At high RH conditions, more water molecules absorbed on the ASS and bacteria, and we found that the other water molecules near to the adsorbed water interact with the adsorbed water via hydrogen bonding, forming a liquid bridge on the joint (Figure S12 (b)). Similarly, a weakly-bound water layer is also found on the spindle knot (Figure S12 (d)), but with a less hydrogen bonds and far distance above the surface. In consist with the structural change, we found that the interaction between water molecules on both the joint and spindle knot are

enhanced to -5.32 and -4.09 eV, respectively, which is contributed to the more hydrogen bonds. From the above discussion, it clearly indicates that at both low RH and high RH, the adhesive force on the joint was stronger than that on the spindle knot. At high RH, the adhesive force was significantly stronger than that at low RH due to more hydrogen bonds.”

Please see SI Page 13 for figure:

Supplementary Fig. 12. Theoretical model of the local hydrogen bonds: at the low (a) and high (b) RH situations on the joint, at the low (c) and high (d) RH situations on the spindle knot. The dark brown, light brown, blue, white and red stick balls are C, Ti, N, H and O atoms, respectively. The dotted green line indicates the hydrogen bonds.

Question 6. *All the Contact Angle data presented is not in the form of average data ($CA \pm ?$, at least three times' repeating), so less the credibility. More seriously, the CAs on the knot is $\sim 126^\circ$, the water advancing and receding contact angle on knot are respectively $\sim 106^\circ$ and 71° , how can contact angle larger than the advancing contact angle? further explanation is required.*

Response: Thanks for your comprehensive comments. In fact, the contact angle was tested for three times in our work, we chose one picture of the contact angle to represent the results, and showed the average contact angle of the tested picture. We are sorry about our misleading mark in the picture. In order to properly explain the results, we have added the average data for three samples in the Fig. 3b.

The difference between the contact angle and advancing contact angle may come from different equipment and test methods. The contact angle in Fig. 3b was conducted by spraying 10 pL water droplet on the ASS immediately (equipment Kruss DSA100 for single fiber water contact angle). The volume of the water droplet is very small and the picture was taken as soon as the droplet contact with the dried ASS, yielding a bigger contact angle. On the other hand, in Fig. 3g, advancing and receding contact angles were *in situ* measured during bioaerosol capture process by OCA15Pro contact angle meter. The droplet *in situ* grew on the ASS with big volume of around 0.5 μ L (much bigger than the 10 pL in Fig. 3b), covering the joint and spindle knot rather than a single area. The big volume of droplet may influence the contact angle due to gravity. In addition, the surface of the ASS has been wetted during bioaerosol capture, resulting in smaller contact angle of advancing and receding contact angles. Therefore, the contact angles in Fig. 3b and Fig. 3g are different in different test situations and equipment. In order to more clearly express the results, we have added the related information in the SI.

Author's changes in Manuscript: Please see Page 29, Line 573

Fig. 3. Bioaerosols capture mechanisms of the ASS photocatalyst. (a) Bioaerosols capture performance of the ASS photocatalyst with different fibre substrate. (b) Water contact angles of the single ASS photocatalyst at different RH. (c) Adhesive force between the bacteria and the ASS photocatalyst at different RH. (d) Optical images of captured bioaerosols by the ASS photocatalyst with different β , height(H) of the spindle knots and length (L) of the joints. (e) Bioaerosols capture performance of the ASS photocatalyst with different morphology. (f) SEM images and AFM images of the ASS photocatalyst. (g) Advancing and receding angles of the droplet on the ASS photocatalyst. (h) Bioaerosols capture performance of the ASS photocatalyst with and without TiO₂. (i) Illustration of the proposed bioaerosols capture and concentration mechanisms of the ASS photocatalyst.

Author's changes in SI: please see Page 4, Line 59-62, “The water contact angle of single fiber at different RH was conducted by a water contact angle tester (Kruss DSA100) equipped with 10 pL water droplet sprayer. The advancing and receding contact angles of the ASS during bioaerosol capture process were *in situ* characterized by

OCA15Pro contact angle meter.”

Question 7. *“ $\theta_A < \theta_R$ means that the droplet prefers to spread towards the spindle knot rather than towards the joints, generating a driving force for droplet movement from the joint to be concentrated onto the spindle knots (described in SI)”, why $\theta_A < \theta_R$ means that? Are there any references? The mechanism of droplet self-propelling should be reconstructed (how surface wettability difference generates directional driving force).*

Response: Thanks for your insightful comments. Sorry for our confused statements. We have deleted the statement of “ $\theta_A < \theta_R$ means that the droplet prefers to spread towards the spindle knot rather than towards the joints” and revised the related statements for accurate expression as following.

Author's changes in Manuscript: Please see Page 12, Line 257-263, “Then, the droplets grow large and make contact with the spindle knot and the joint of the ASS photocatalyst, forming θ_A (advancing angle) and θ_R (receding angle), respectively. As shown in Fig. 3i (III), the spindle knot is composed of TiO_2 nanoparticles, while the joint is smooth nylon fiber, therefore, the spindle knot region has a higher roughness than joint region, which give rises to a driving force generated by a surface energy gradient to move the droplet towards the spindle knot...”.

Reviewer #3 (Remarks to the Author):

Comments: *This manuscript reports the photocatalytic bacterial inactivation of a bioinspired artificial spider silk (ASS) made from a periodic spindle structure of TiO₂ on a nylon fiber. ASS showed the capture capacity of the bioaerosols twice as high as the pure nylon fiber and 99.99% inactivation efficiency on both Gram-negative E. coli bacteria and Gram-positive B. subtilis bacteria under UV light. The authors attributed its capture mechanism to the synergistic effect from the hydrophilicity, size of the spindle knots, and surface roughness. The concentrated bacteria on the spindle knots are inactivated by ROS generated through the photocatalytic reactions. The results provide a very interesting strategy that can be adopted for the purification system of the bioaerosols without using the high-power air exhaust system.*

Response: Many thanks for your highly positive to our work. We have further improved our manuscript according to reviewer's comments.

Question 1. *The authors termed the hydrophobicity as a water contact angle (θ) $> 90^\circ$ and the hydrophilicity as that $< 90^\circ$, so the surface with $\theta=95.8^\circ$ was hydrophobic while that with $\theta=87.6^\circ$ was hydrophilic. But both seem to be hydrophobic to me. Such a criterion may mislead the surface characteristics.*

Response: It is widely accepted that a surface is hydrophobic when its static water contact angle θ is $>90^\circ$ and is hydrophilic when θ is $<90^\circ$ ^{23,44}. Therefore, $\theta=87.6^\circ$ is hydrophilic. The results indicates that the hydrophilicity of the ASS is improved at high RH. In order to clearly clarify the results, we have cited the references and revised the statements in the manuscript.

Author's changes in Manuscript: Please see Page 9, Line 182-186. “Furthermore, Fig. 3b shows that the water contact angle (θ) of the joint at RH 50% was 97.5° ($\theta>90^\circ$, hydrophobic), while at RH 80%, it was 88.9° ($\theta<90^\circ$, hydrophilic)^{23,44}. For the spindle knots, θ was 125.3° at RH 50% and decreased to 93.6° at RH 80%. The results indicates that the hydrophilicity of the ASS is improved at high RH.”.

Reference: Ref. 23. Bai, H. et al. Direction controlled driving of tiny water drops on bioinspired artificial spider silks. *Adv. Mater.* **22**, 5521-5525 (2010);

Ref. 44. Law, K. Y. Definitions for hydrophilicity, hydrophobicity, and superhydrophobicity: getting the basics right. *J. Phys. Chem. Lett.* **5**(4), 686-688(2014).

Question 2. Also, they used AFM with a bacterial probe to measure its adhesion with the surface of the spindle knots and the joints. But they did not identify the type of the bacteria used for an AFM probe. Isn't the surface interaction dependent on the type of the bacteria on the AFM probe?

Response: In order to figure out if the types of bacteria used as AFM probe influence the interaction, we newly supplied the test of *B. subtilis* (gram-positive) as the AFM probe, and found the similar results with *E. coli* (gram-negative) probe as follows. It can be seen that adhesive forces between bacteria and joints are higher than with spindle knots. The adhesive forces were all enhanced at high RH. The type of the bacteria used as AFM probe has been marked in the Fig. 3c and Supplementary Fig. 11. The representative adhesive forces for *B. subtilis* also added in Supplementary Fig. 13.

Author's changes in Manuscript: Please see Page 9-10, Line 192-196: “As shown in the bottom of Fig. 3c, the average adhesive forces between the *E. coli* bacteria and the surface of the spindle knots and joints were 8.4 and 9.0 nN for RH 50% and 25.3 and 28.1 nN for RH 80%, respectively. For *B. subtilis*, the adhesive forces of the spindle knot and the joint were 23.3 and 27.6 nN for RH 50%, and 32.6 and 40.4 nN for RH 80%, respectively.”.

Page 29, Line 573 for figure:

Fig. 3. Bioaerosols capture mechanisms of the ASS photocatalyst. (c) Adhesive force between the bacteria and the ASS photocatalyst at different RH.

Author's changes in SI:

Supplementary Fig. 11. Optical images of bacterial probes for AMF characterization. *E. coli* bacterial probe at (a) low and (b) high magnification, and (c) *B. subtilis* bacterial probe

Supplementary Fig. 13. Representative adhesive forces between the (a) *E. coli* and (b) *B. subtilis* bacterial probe and the ASS photocatalyst at different RH

Question 3. *There have been quite a few reports on ASS for its design and structure with potential applications, but its use in the photocatalytic purification process is an interesting idea by creating the periodic spindle structure of well-known photocatalytic oxide (TiO_2). While the structural design and processing scheme seem to be well executed, some explanations on the resultant microstructure details are missing and bacteria killing mechanisms are not clear. For example, it creates the periodic fiber spindle knots made of the TiO_2 – PMMA cluster, but detailed representation of such spindle knot surfaces was not given in the manuscript so it is not clear how surface morphologies play a role in the effective photocatalytic inactivation of bacteria. TiO_2 nanoparticles need to be exposed to the surface to be effective, so microstructure characterization of the hybrid arrangements of TiO_2 in PMMA resulting from its various concentrations should be discussed in relation to the corresponding antibacterial properties. In Fig. 4 (d), the bacterial inactivation efficiency is shown to increase with TiO_2 loadings (i.e., more ROS production) until it reaches to a certain concentration of TiO_2 . In order to relate the origin of the saturation, microstructure evolution at the top surface of the TiO_2 – PMMA mixture needs to be explained. Degussa P25 is well-known TiO_2 nanoparticles, but it still needs to report the range of particle sizes of TiO_2 in*

the manuscript.

Response: Thanks for the precious comments. We have added the surface morphology of the ASS with different TiO₂ concentrations at high magnifications and particle size distribution of P25 in the SI.

We have added the detailed surface morphology of the ASS with different TiO₂ concentration in Supplementary Fig. 4. It can be seen that with increase of TiO₂ concentration in the PMMA solution, more TiO₂ nanoparticles expose on the surface. As discussed in the manuscript (Fig. 4d), the inactivation efficiency of the ASS can be influenced by the TiO₂ loading. However, the balance between the structure of the ASS and inactivation efficiency should be considered. Therefore, we majorly investigated the ratio of TiO₂/PMMA of 1:100 in the study.

Particle size distribution of P25 is shown in Supplementary Fig. 17.

Author's changes in Manuscript: Please see Page 6, Line 110-111 “With increase of TiO₂ concentration, more TiO₂ nanoparticles exposed on the surface (Supplementary Fig. 4).”

Please see Page 18, Line 387-389 “Nylon fibres, titanium dioxide (P25, Supplementary Fig. 17.) and polymethyl methacrylate (PMMA) were purchased from Macklin Co., Ltd., and the other materials used in this study are listed in the Supporting Information (SI).”

Author's changes in SI: Please see Page 8, Line 135

Supplementary Fig. 4 Surface morphology of the ASS with different TiO₂ concentrations in PMMA
 Please see Page 15, Line 215: Particle size distribution of P25 is shown in Supplementary Fig. 17.

Supplementary Fig. 17. Particle size distribution of P25 TiO₂ nanoparticles

Question 4. *Bacterial inactivation mechanism needs some clarification too. The authors stated that the cell wall of the Gram-negative bacteria (*E. coli*) is thinner than that of the Gram-positive bacteria (*B. subtilis*), making the former more easily inactivated by ROS. However, the Gram-negative bacteria has an additional outer membrane (which is not present for the Gram-positive bacteria), so they are in general more resistant to anti-biotics and causing more serious diseases for human. They may be more difficult to inactivate with the ROS generated from TiO₂ photocatalysts as well. Furthermore, the author's data do not warrant their claim either. Therefore, it may not be attributed to the blockage of light absorption due to more *E. coli* bacteria on fibers.*

Response: Thanks for your instructive comments. Recent studies have overwhelmingly evidenced that the cell inactivation by photocatalyst is due to the attack of ROSs, leading to membrane and cell wall damage. The Gram-negative bacteria (*E. coli*) have a complex cell wall and additional outer membrane-lipopolysaccharide, comparing with the Gram-positive bacteria (*B. subtilis*), which protect themselves from the attack of ROSs to a certain extent. Thus, we agree with the reviewers' comment and revised the related statement in the manuscript.

Author's changes in Manuscript: Please see Page 15, Line 315-321, “The C/C₀ of *B. subtilis* is slightly smaller than that of *E. coli*, indicating *B. subtilis* can be inactivated more easily. Recent studies have evidenced that the cell inactivation by photocatalyst is due to the attack of ROSs, leading to membrane and cell wall damage³⁷. The Gram-negative bacteria (*E. coli*) have a complex cell wall and additional outer membrane-lipopolysaccharide, comparing with the Gram-positive bacteria (*B. subtilis*). Thus, Gram-negative bacteria (*E. coli*) can protect themselves from the attack of ROSs to a certain extent^{46,47}.”.

References: Ref. 37. Wang, W. et al. Photocatalytic nanomaterials for solar-driven bacterial inactivation: recent progress and challenges. *Environ. Sci. Nano*, **4**,782-799 (2017);

Ref. 46. Dahl, T. A., Midden, W. R., & Hartman, P. E. Comparison of killing of gram-negative and gram-positive bacteria by pure singlet oxygen. *J. Bacteriol.* **171**(4), 2188-2194 (1989);

Ref. 47. He, J., Zheng, Z., & Lo, I. M. Different responses of gram-negative and gram-positive bacteria to photocatalytic disinfection using solar-light-driven magnetic TiO₂-based material, and disinfection of real sewage. *Water Research*, **207**, 117816 (2021).

Question 5. *The testing results need to be compared with the control sample, but this study provides no data from the bacterial viability testing with the pure nylon fiber or the ASS catalysts without TiO₂. This is because these bacteria can be killed under UV light or after the droplets dry up on the control sample surfaces. Further, in the Results and Conclusion sections, the author claimed 99.99% efficiency (4 log reduction) while it was 99.999% (5 log reduction) in the Abstract. This discrepancy needs to be corrected.*

Response: Thanks for your precise comments. The photocatalytic inactivation performance of control sample surfaces without TiO₂ has been compared and is shown in Fig. 4d (sample with 0% TiO₂).

Thanks also give to your kind reminder, and we have deleted the inactivation efficiency of “99.999%” and revised to “99.99%” in the abstract. The results have been unified in abstract, discussion and conclusion sections accordingly in the manuscript.

Fig. 4d Inactivation efficiency of the ASS photocatalyst with different TiO₂ loading

Question 6. *Figure 5 (d) is confusing. It was supposed to show that more ROS were generated at the surface of ASS in air or in the water droplet captured on ASS, but the two*

highest intensity cases were 'ASS/luminol' and 'Luminol irradiated'. I am not sure how the label of these samples is related to the aforementioned two cases (at the surface of ASS in air; inside the water droplet with the ASS photocatalyst).

Response: Thanks for your question. Sorry for our confusing labels, and we have revised the labels in Fig. 5d as suggested.

The spectrum of substrate with UV irradiation for 5 min was labeled as "Substrate+UV"; The spectrum of the ASS without UV pre-irradiation was labeled as "ASS"; The spectrum of the ASS with UV irradiation in water interface for 5 min was labeled as "ASS+UV+water"; The spectrum of luminol with UV irradiation was labeled as "Luminol+UV"; The spectrum of the ASS adding luminol with UV irradiation was labeled as "ASS+luminol+UV".

Author's changes in Manuscript: Please see Page 31, Line 592,

Fig. 5. Bioaerosols inactivation mechanisms of the ASS photocatalyst. (d) Chemiluminescence spectra of luminol dropping on the ASS photocatalyst before and after light irradiation, the ASS photocatalyst in interface and water, and comparisons of the luminol, the ASS photocatalyst and the combination.

Question 7. Photocatalytic inactivation properties of ASS were measured under the UV light with a wavelength of 365 nm. But the ROS generation experiments were done with the UV light with a wavelength of 325 nm. This discrepancy needs to be explained to make the relevance between two experiments.

Response: Thanks for the nice question. The wavelengths of commonly used LED UV lamps are 254 nm and 365 nm. The wavelength of <380 nm can be absorbed by TiO₂ for photocatalytic reaction. Therefore, both lamps can be used in photocatalytic

inactivation. However, UV light at 254 nm is harmful to humans and is usually used in UV disinfection, as it damages DNA in cells significantly. UV light at 365 nm shows better penetration and less harmful to humans, and also can be strongly absorbed by TiO₂ for photocatalytic reaction. Therefore, UV light at 365 nm is usually used in photocatalytic inactivation experiment

On the other hand, the wavelengths of the laser microscopic confocal Raman spectrometer (LabRAM HR Evolution) are 325 nm, 532 nm, 633 nm, 785 nm and 1064 nm. The laser wavelength of 325 nm used in Raman equipment is closest to that used in photocatalytic inactivation experiment (365 nm). Therefore, the wavelength of 325 nm UV light for ROS generation experiment was used. To explain this discrepancy to make the relevance between two experiments, we have added some explanations in the **SI** as suggested.

Author's changes in SI: Please see Page 5, Line 94-97, “The laser wavelength of 325 nm used in Raman equipment is closest to that used in photocatalytic inactivation experiment (365 nm). Therefore, the wavelength of 325 nm UV light for ROS generation experiment was used.”

Question 8. *In the References section, only a single author is included in each of the reference article. Normally, full author names should be included in the references.*

Response: Thanks for your nice suggestion. We have revised the format of the references referring to the format of *Nature Communications* published paper. The only a single author is listed when there are more than five authors.

Question 9. *In summary, this paper presents a novel approach to mimic a spider silk by using a periodic spindle structure of TiO₂ on a nylon fiber. The processing schemes are well designed to capture the bioaerosols that contain bacteria. But there seemed to be no clear microstructural description of TiO₂ nanoparticles spread in PMMA and their influence on antibacterial performances under UV light. In addition, some of the experimental techniques are not well coordinated each other, so it requires more explanation. Also, there were no data provided from control sample*

for the bacterial viability test. Inactivation mechanism of the bacteria via photocatalytic reactions needs to be better explained or confirmed with the data to support the author's conclusion on the Gram-negative vs. the Gram-positive bacteria.

Response: Thanks for the precious suggestions. The microstructural description of TiO₂ and influence on inactivation performance of the ASS photocatalyst were added and discussed as suggested in Supplementary Fig. 4 and Fig. 4d, respectively. The detailed response can be found in your **Question 3**.

The experimental techniques and setups were explained in the SI. The detailed response can be found in your **Question 7**.

The control sample for inactivation performance of the ASS photocatalyst was also shown in Fig. 4d with 0% TiO₂ loading. The detailed response can be found in your **Question 5**.

The related mechanisms of inactivation mechanism are discussed and cited in the manuscript. The detailed response can be found in your **Question 4**.

Finally, we have revised the manuscript accordingly and the point-to-point answers are presented in above questions.

REVIEWERS' COMMENTS

Reviewer #1 (Remarks to the Author):

The authors made improvements following the reviewers' comments. The manuscript language has to be thoroughly checked before its publication. Presently, there are many grammar errors throughout the manuscript text, e.g., verb tense, "is consisted", et al. For the manuscript title, usually people do not say this : bacteria in bioaerosol. It should be Bacteria Aerosol. The manuscript abstract should highlight the new idea, not just the efficiency. 99.99% inactivation efficiency alone is not attractive enough. Additionally, the inactivation time required should be also presented explicitly in the abstract.

Reviewer #2 (Remarks to the Author):

Authors have made detailed explanation in response to the questions raised by reviewers. Supplementary experiments are carried out and conclusions are convictive. The revision of manuscript can be recommended for publication.

Reviewer #3 (Remarks to the Author):

Authors responded well to the review comments and questions. The revision is acceptable to this reviewer.

Response to the comments for NCOMMS-22-48373A

Title: Bioinspired Artificial Spider Silk Photocatalyst for the High-efficiency Capture and Inactivation of Bacteria Aerosols

REVIEWER COMMENTS

Reviewer #1 (Remarks to the Author):

Comments: The authors made improvements following the reviewers' comments. The manuscript language has to be thoroughly checked before its publication. Presently, there are many grammar errors throughout the manuscript text, e.g., verb tense, "is consisted", et al. For the manuscript title, usually people do not say this : bacteria in bioaerosol. It should be Bacteria Aerosol. The manuscript abstract should highlight the new idea, not just the efficiency. 99.99% inactivation efficiency alone is not attractive enough. Additionally, the inactivation time required should be also presented explicitly in the abstract.

Response: Thank for your comments, and we have revised the grammar errors throughout the manuscript. The manuscript title has been revised to “Bioinspired Artificial Spider Silk Photocatalyst for the High-efficiency Capture and Inactivation of Bacteria Aerosols”. In addition, we have revised the abstract according to the comments.

Authors' change in the manuscript: Page 1, Line 7, “Bioaerosol can cause the spread of disease, and therefore, capture and inactivation of bioaerosols is desirable. However, filtration systems can easily become blocked, and are often unable to inactivate the bioaerosol once it is captured. Herein, we reported a bioinspired artificial spider silk (ASS) photocatalyst, consisting of a periodic spindle structure of TiO₂ on nylon fibre that can efficiently capture and concentrate airborne bacteria, followed by photocatalytic inactivation in situ, without a power-supply exhaust system. The ASS photocatalyst exhibits a higher capture capacity than the nylon fibre substrate and a photocatalytic inactivation efficiency of 99.99% obtained

under 4 h irradiation. We found that the capture capacity of the ASS photocatalyst can be mainly attributed to the synergistic effects of hydrophilicity, Laplace pressure differences caused by the size of the spindle knots and surface energy gradients induced by surface roughness. The bacteria captured by the ASS photocatalyst are inactivated by photocatalysis within droplets or at the air/photocatalyst interfaces. This strategy paves the way for constructing materials for bioaerosol purification.”

Reviewer #2 (Remarks to the Author):

Comments: Authors have made detailed explanation in response to the questions raised by reviewers. Supplementary experiments are carried out and conclusions are convictive. The revision of manuscript can be recommended for publication.

Response: Many thanks for your highly positive to our work. We have further improved our manuscript according to reviewer’s comments.

Reviewer #3 (Remarks to the Author):

Comments: Authors responded well to the review comments and questions. The revision is acceptable to this reviewer.

Response: Many thanks for your highly positive to our work. We have further improved our manuscript according to reviewer’s comments.